# ActFusion: a Unified Diffusion Model for Action Segmentation and Anticipation

**Dayoung Gong**      **Suha Kwak**      **Minsu Cho**
Pohang University of Science and Technology (POSTECH)
{dayoung.gong, suha.kwak, mscho}@postech.ac.kr

## Abstract

Temporal action segmentation and long-term action anticipation are two popular vision tasks for the temporal analysis of actions in videos. Despite apparent relevance and potential complementarity, these two problems have been investigated as separate and distinct tasks. In this work, we tackle these two problems, action segmentation and action anticipation, jointly using a unified diffusion model dubbed ActFusion. The key idea to unification is to train the model to effectively handle both visible and invisible parts of the sequence in an integrated manner; the visible part is for temporal segmentation, and the invisible part is for future anticipation. To this end, we introduce a new anticipative masking strategy during training in which a late part of the video frames is masked as invisible, and learnable tokens replace these frames to learn to predict the invisible future. Experimental results demonstrate the bi-directional benefits between action segmentation and anticipation. ActFusion achieves the state-of-the-art performance across the standard benchmarks of 50 Salads, Breakfast, and GTEA, outperforming task-specific models in both of the two tasks with a single unified model through joint learning.

## 1   Introduction

In everyday life, when interacting with people, we anticipate their future actions while recognizing their actions observed in the present and the past. Similarly, for effective human-robot interaction, robotic agents must recognize ongoing actions while anticipating future behaviors. Two essential tasks in computer vision for such a temporal understanding of human actions are temporal action segmentation (TAS) [38, 19, 32, 61, 12, 65, 3, 43, 63, 8] and long-term action anticipation (LTA) [2, 34, 25, 45]. The task of TAS aims at translating observed video frames into a sequence of action segments, while the goal of LTA is to predict a plausible sequence of actions in the future based on the observed video frames. These tasks are closely related in terms of understanding the relations between actions; recognizing actions in the present and the past may improve anticipating action in the future, and the ability to anticipate the future may also enhance recognizing observable actions when facing visual ambiguities.

Despite the apparent relevance and potential complementarity, these two problems have been investigated as separate and distinct tasks. While a growing body of work has shown remarkable progress on both TAS and LTA, these methods are primarily designed for one task (see Fig. 1a) and show poor generalization when applied to the other (e.g., see FUTR [25] and DiffAct [43] in Fig. 1c). Some prior methods tackle both tasks simultaneously, but they rely on task-specific architectures and require separate training for each individual task (e.g., see TempAgg [52] in Fig. 1c).

In this paper, we introduce ActFusion, a unified diffusion model that addresses TAS and LTA through a single architecture and training process, as illustrated in Fig. 1b. The core idea of our unification lies in training the model to effectively handle two different parts of the sequence: visible parts for action segmentation and invisible parts for action anticipation. Accordingly, we propose a

38th Conference on Neural Information Processing Systems (NeurIPS 2024).

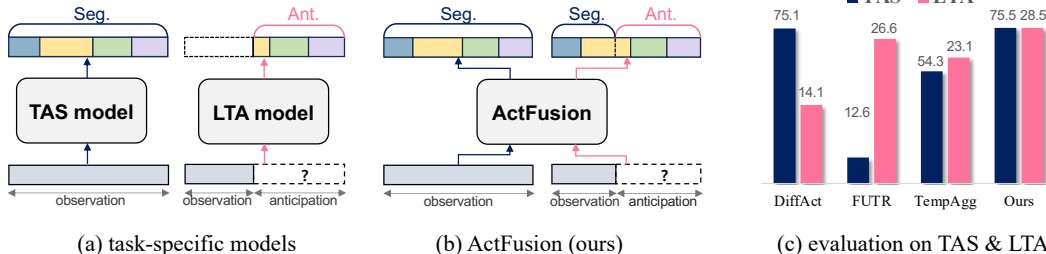

|  | (a) task-specific models | (b) ActFusion (ours) | (c) evaluation on TAS & LTA |

Figure 1: **Task-specific models vs. ActFusion (ours).** (a) Conventional task-specific models for TAS and LTA. (b) Our unified model ActFusion to address both tasks. (c) Performance comparison across tasks. Tasks-specific models such as DiffAct [43] for TAS and FUTR [25] for LTA exhibits poor performance on cross-task evaluations. ActFusion outperforms task-specific models on both TAS and LTA, including TempAgg [52], which trains separate models for each task. Note that the performance of ActFusion is evaluation result of a single model through a single training process. The reported performance represents the average of each task in the original paper or evaluated with the official checkpoint (See Sec. G for details).

new anticipative masking strategy in which a late part of the video frames is masked as invisible, and learnable tokens replace it. We also introduce a random masking strategy to help the model accurately identify actions when some parts of the video are ambiguous [59], drawing inspiration from previous methods [15, 6, 28, 59]. ActFusion generates action sequences from Gaussian noise through iterative denoising, conditioned on both visual features and mask tokens. Through this process, the model jointly learns to classify action labels of visual features and predict future actions of mask tokens. As shown in Fig. 1c, ActFusion achieves superior performance in both action segmentation and anticipation, demonstrating stronger cross-task generalization compared to previous methods [43, 25, 52].

Furthermore, we find that the benchmark evaluation of some prior LTA methods [2, 20, 25] has exploited the ground-truth length of input videos in testing time, which makes the evaluation unrealistic and problematic since the duration of future actions is supposed to be unknown in advance. To address this issue, we conduct comprehensive experiments both with and without ground-truth length information, providing insights into more realistic deployment scenarios.

Our contribution can be summarized as follows: **1)** We introduce ActFusion, a unified diffusion model that jointly addresses TAS and LTA through a single training process. **2)** We present anticipative masking for effective task unification, along with random masking to enhance robustness against visual ambiguities. **3)** Comprehensive experimental results show that jointly learning both tasks provides bi-directional benefits. ActFusion achieves state-of-the-art performance on TAS and LTA across benchmark datasets - 50 Salads and Breakfast, and GTEA - demonstrating the effectiveness of the proposed method.

## 2 Related work

**Temporal action segmentation (TAS).** The goal of TAS [19, 65, 43, 63, 12, 23, 5, 8, 32, 3, 16] is to classify frame-wise action labels in a video. Earlier approaches employ temporal sliding windows [50, 33] for action segment detection, grammar-based methods [37, 36] have been introduced to incorporate a temporal hierarchy of actions during segmentation. Temporal convolution networks [38, 19] and Transformer-based models [65, 8] are introduced based on deep learning methods. Since learning long-term relations of actions from activity videos is challenging, a series of work has been proposed to develop refinement strategies [30, 32, 23, 12, 57, 63, 3, 24] that can be applied to the TAS models [19, 65]. Recently, DiffAct [43] is proposed to iteratively denoising action predictions conditioned on the input video features adopting the diffusion process. In a similar spirit, the proposed ActFusion is based on the diffusion process, focusing on unifying TAS and LTA through anticipative masking. We demonstrate the bidirectional benefits between the two tasks, showing that learning segmentation along with anticipation is effective.

**Long-term action anticipation (LTA).** LTA [2, 1, 34, 20, 52, 25, 44, 26] has recently emerged as a crucial task for predicting a sequence of future actions in long-term videos. Initial models use RNNs

and CNNs [2], while time-conditioned anticipation [34] introduces one-shot anticipation of specific future timestamps. A GRU-based model [20] used cycle-consistent relations of past and future actions. Sener et al. [52] proposed TempAgg, a multi-granular temporal aggregation method for action anticipation and recognition, utilizing different model architectures and task-specific losses for the two tasks. Gong *et al.* [25] propose a transformer model for parallel anticipation, dubbed FUTR, empirically showing that learning action segmentation as an auxiliary task is helpful for anticipation. Nawhal *et al.* [44] propose a two-stage learning approach for LTA and Zhang *et al.* [66] present object-centric representations using visual-language pre-trained models for LTA. While previous work [52, 20, 25, 44] adopts TAS as an auxiliary task to help learn LTA, a unified model evaluated on both tasks is rarely explored, often showing poor cross-task generalization performance (Fig. 1c). The exception is TempAgg, which requires separate training and model architecture for the two tasks. In this work, we present a unified model that tackles both TAS and LTA in a single training process.

**Diffusion models.** Recent success in denoising diffusion models [55, 29, 56] opens a new era of computer vision research. The diffusion models learn the original data distribution through the iterative denoising process. Diffusion models have recently shown notable success in various domains, such as image generation [51, 14], video generation [42, 27, 64], object detection [13], semantic segmentation [7], temporal action segmentation [43], self-supervised learning [62], and etc [18]. Recently, DiffMAE [62] integrates masked autoencoders with the diffusion models, where the model learns to denoise masked input while learning data distributions through generative pre-training of visual representations. In this work, we present a unified diffusion model that effectively integrates TAS and LTA through masking, where the model learns to denoise action labels conditioned on both visual and mask tokens. In this way, the model can effectively learn temporal relations between actions by classifying visual tokens and inferring missing actions of the mask tokens.

# 3 Preliminary

In this section, we give a brief overview of diffusion models [29, 54]. The diffusion models learn a data distribution by mapping and denoising noises from the original data distribution. The training process of diffusion models involves forward and reverse processes from random noise. During the forward process, Gaussian noise is added to the original data, while in the reverse process, a neural network learns to reconstruct the original data by iteratively removing noise.

The forward process, or diffusion process, transforms the original data $x^0$ into noisy data $x^s$:

$$x^s = \sqrt{\gamma(s)}x^0 + \sqrt{1 - \gamma(s)}\epsilon. \tag{1}$$

Here, a noise $\epsilon \sim N(0, \mathbf{I})$ is added to the original data distribution $x^0$ following the decreasing function $\gamma(s)$ of time step $s \in \{1, 2, ..., S\}$, where $S$ represents the entire forward time-step. Note that the function $\gamma(s)$ determines the intensity of the noise added to the original data following the pre-defined variance schedule [29].

In the reverse process, a neural network $f(x^s, s)$ is trained to recover the original data $x^0$ from noisy data $x^s$ using $l_2$ regression loss:

$$\mathcal{L} = \frac{1}{2}||f(x^s, s) - x^0||^2. \tag{2}$$

During inference, the model $f$ iteratively denoises pure noise $x^S$ to reconstruct the original data $x^0$, *i.e.*, $x^S \rightarrow x^{S-\Delta} \rightarrow ... \rightarrow x^0$, following the updating rule [29, 54]. We refer the reader to [29, 54] and Sec. A for more details.

In our context, the neural network learns to generate action sequences from Gaussian noise, conditioned on visible features for action segmentation and mask tokens for action anticipation.

# 4 Proposed approach

We present ActFusion, a unified diffusion model for action segmentation and anticipation. This section describes the problem setup in Sec. 4.1, the model architecture in Sec. 4.2, and the proposed masking strategies and training objectives in Sec. 4.3 and Sec. 4.4, respectively.

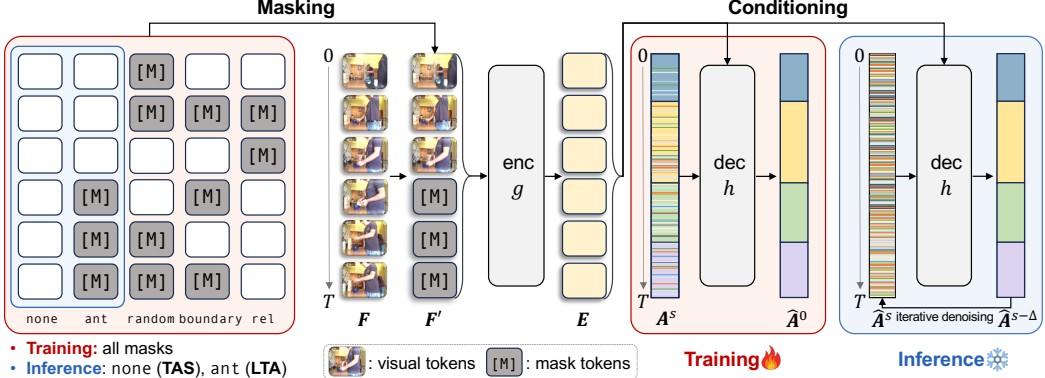

Figure 2: **Overall pipeline of ActFusion.** During training, we randomly select one of five masking strategies and apply it to input video frames $\boldsymbol{F}$, replacing masked regions with learnable tokens to obtain masked features $\boldsymbol{F}'$. These features are processed by the encoder $g$ to produce visual embeddings $\boldsymbol{E}$, which condition the decoder $h$ to denoise action labels from $\boldsymbol{A}^s$ to $\hat{\boldsymbol{A}}^0$ at time-step $s$. For inference, we use different masking strategies depending on the task: no masking for TAS and anticipative masking for LTA. The decoder then iteratively denoises action labels following $\hat{\boldsymbol{A}}^S \rightarrow \hat{\boldsymbol{A}}^{S-\Delta} \rightarrow ... \rightarrow \hat{\boldsymbol{A}}^0$ using the DDIM update rule [54].

## 4.1 Problem setup

Temporal action segmentation (TAS) aims to classify input video frames into a sequence of predefined action classes, while long-term action anticipation (LTA) predicts future actions based on partially observed video sequences. Formally, given a video sequence $\boldsymbol{F} = [F_1, F_2, \cdots, F_T]$ of length $T$, TAS predicts frame-wise action labels $\boldsymbol{A} = [A_1, A_2, \cdots, A_T]$, where each $A_i$ is a one-hot vector representing the action class. In LTA, given the first $N^O = \lceil \alpha T \rceil$ observed frames, the goal is to anticipate action labels for the subsequent $N^A = \lceil \beta T \rceil$ frames, where $\alpha \in [0, 1]$ and $\beta \in [0, 1 - \alpha]$ represent the observation and anticipation ratios, respectively. Here, $\lceil \cdot \rceil$ denotes the ceiling function.

## 4.2 ActFusion

ActFusion aims to unify TAS and LTA by leveraging an encoder-decoder architecture with adaptive masking strategies. Figure 2 illustrates the overall pipeline, where ActFusion consists of a masked encoder $g$ and a denoising decoder $h$. The encoder obtains visual features and mask tokens as input and generates embedded tokens as output. The decoder then progressively reduces noises through an iterative denoising process conditioned on the embedded tokens.

During training, we randomly sample a time step $s \in \{1, 2, ..., S\}$ for each iteration and add noise $\epsilon \sim N(0, \mathbf{I})$ to the ground-truth action labels $\boldsymbol{A}^0$ following Eq. 1, resulting in noisy action labels $\boldsymbol{A}^s$. The decoder then aims to denoise $\boldsymbol{A}^s$ to reconstruct the original action labels $\hat{\boldsymbol{A}}^0$. During inference, the decoder starts with Gaussian noise $\hat{\boldsymbol{A}}^S$ and progressively denoises it following the DDIM [54] update rule to generate final predictions $\hat{\boldsymbol{A}}^0$, *i.e.*, $\hat{\boldsymbol{A}}^S \rightarrow \hat{\boldsymbol{A}}^{S-\Delta} \rightarrow ... \rightarrow \hat{\boldsymbol{A}}^0$.

The key to our unified approach lies in training the model to effectively process both visible and invisible parts of the sequence, where the visible part corresponds to observed video frames and the invisible part represents future frames to be anticipated. To this end, we introduce anticipative masking that replaces unobserved frames with learnable mask tokens, enabling the model to learn future predictions. We apply this masking strategy consistently during both training and inference to achieve joint learning of action segmentation and anticipation. We further incorporate random masking, where video frames are randomly masked to enhance robustness against visual ambiguities. Both masking strategies are detailed in Sec. 4.3.

**Input structuring.** Given input video features $\boldsymbol{F} \in \mathbb{R}^{T \times C}$ with $T$ frames of $C$ feature dimensions, we start by defining a binary mask matrix $\boldsymbol{M} \in \{0, 1\}^{T \times 1}$. This mask serves as a frame selector, where a value of 0 indicates a frame to be masked and replaced by mask tokens, while a value of 1 indicates an unmasked frame. The learnable mask token, denoted as $\boldsymbol{m} \in \mathbb{R}^{1 \times C}$, replaces the visual

features in frames selected for masking, producing the input features $\boldsymbol{F}'$ for the encoder:

$$\boldsymbol{F}' = \boldsymbol{F} \odot \boldsymbol{M} + (\boldsymbol{1}_{T \times C} - \boldsymbol{M}) \odot \boldsymbol{m}, \qquad (3)$$

where $\odot$ denotes element-wise multiplication and $\boldsymbol{1}_{i \times j}$ represents a matrix of ones with dimensions $i \times j$. Here, $\boldsymbol{M}$ and $\boldsymbol{m}$ are broadcasted along the channel and temporal dimensions, respectively.

For our model architecture, we use a modified version of ASFormer [65] used in DiffAct [43], which employs dilated attention to capture both local and global relations in the input sequence (See Fig. S2 for the detailed model architecture).

**Encoder.** The encoder consists of the $N^{\mathrm{E}}$ layers, each consisting of a dilated 1-d convolution followed by instance normalization, dilated attention, and a feed-forward network [65]. In the dilated attention mechanism, the receptive field is limited to a local window size $w = 2^i$ for the $i$-th layer, where the increasing dilation captures progressively broader temporal relations. The output of each layer is combined with its input via a residual connection before proceeding to the subsequent layer. Given the input features $\boldsymbol{F}'$, the encoder $g$ produces embedded tokens $\boldsymbol{E} \in \mathbb{R}^{T \times D}$ as:

$$\boldsymbol{E} = g(\boldsymbol{F}'), \qquad (4)$$

where $D$ represents the dimensions of embedded tokens. A fully-connected (FC) layer $\boldsymbol{W}^{\mathrm{enc}} \in \mathbb{R}^{D \times K}$ is applied to $\boldsymbol{E}$ to obtain frame-wise classification logits from the encoder followed by a softmax:

$$\hat{\boldsymbol{A}}^{\mathrm{enc}} = \sigma(\boldsymbol{E}\boldsymbol{W}^{\mathrm{enc}}), \qquad (5)$$

where $K$ is the number of action classes and $\sigma$ is the softmax.

**Decoder.** The decoder is composed of $N^{\mathrm{D}}$ sequential layers. Each layer consists of dilated 1-d convolution, dilated attention, instance normalization, and feed-forward networks. As in the encoder, the dilation ratio for the $i$-th layer is set to $w = 2^i$. The output of each layer is combined with its input through a residual connection before being passed into the subsequent layer. Given a time step $s$, noisy action labels $\boldsymbol{A}^s$ at time step $s$, and encoder embeddings $\boldsymbol{E}$, the decoder $h$ produces output embeddings $\boldsymbol{Y} \in \mathbb{R}^{T \times D}$ according to:

$$\boldsymbol{Y} = h(\boldsymbol{A}^s, s, \boldsymbol{E}). \qquad (6)$$

The final action label predictions $\hat{\boldsymbol{A}}^0 \in \mathbb{R}^{T \times K}$ are obtained by projecting $\boldsymbol{Y}$ through a fully-connected layer $\boldsymbol{W}^{\mathrm{dec}} \in \mathbb{R}^{D \times K}$ followed by softmax:

$$\hat{\boldsymbol{A}}^0 = \sigma(\boldsymbol{Y}\boldsymbol{W}^{\mathrm{dec}}). \qquad (7)$$

### 4.3 Masking strategy

We introduce two distinct masking strategies: anticipative masking and random masking.

**Anticipative masking.** Anticipative masking enables joint learning of TAS and LTA by separating visible and invisible parts of the input sequence. This strategy masks the latter portion of video frames, requiring the model to predict future actions based on observed frames. Given a video length $T$, we define a binary anticipation mask $\boldsymbol{M}^{\mathrm{A}} \in \{0,1\}^T$ that sets visible frames to 1 and invisible frames to 0: $\boldsymbol{M}_i^{\mathrm{A}} = \mathbb{1}(i \leq N^{\mathrm{O}})$, where $\mathbb{1}$ is an indicator function and $N^{\mathrm{o}}$ represents the number of observed frames. Unlike causal masking [60] that prevents attending to future tokens, our anticipative masking allows interactions among visible tokens while maintaining a clear boundary for anticipation.

**Random masking.** Random masking aims to provide robustness in prediction when parts of video frames are missing or ambiguous [59]. A video is first divided into pre-defined clips of size $Q$, resulting in $N^{\mathrm{P}} = \lceil \frac{T}{Q} \rceil$ total clips. Then, $N^{\mathrm{R}}$ clips are randomly selected to be masked. A binary random mask is defined by $\boldsymbol{M}_i^{\mathrm{R}} = \mathbb{1}(\exists j \in \mathbb{P}, (j-1)Q < i \leq jQ)$, where $\mathbb{P}$ is a randomly selected subset of $\{1, \cdots, N^{\mathrm{P}}\}$ with $|\mathbb{P}| = N^{\mathrm{R}}$.

For fully observable scenarios, we utilize a no mask strategy $\boldsymbol{M}^{\mathrm{N}}$ where all frames remain visible. Additionally, we adopt two masking strategies [43], the relation mask $\boldsymbol{M}^{\mathrm{S}}$ and the boundary mask $\boldsymbol{M}^{\mathrm{B}}$, as illustrated in Fig. 2. The relation mask $\boldsymbol{M}^{\mathrm{S}}$ randomly masks segments associated with an action class to learn inter-action dependencies. The boundary mask $\boldsymbol{M}^{\mathrm{B}}$ masks frames at action transitions to enhance boundary detection. See Sec. C for the detailed formulations of the masks.

During training, one of five masking strategies is randomly selected: no mask $\boldsymbol{M}^{\mathrm{N}}$, anticipative mask $\boldsymbol{M}^{\mathrm{A}}$, random mask $\boldsymbol{M}^{\mathrm{R}}$, relation mask $\boldsymbol{M}^{\mathrm{S}}$, and boundary mask $\boldsymbol{M}^{\mathrm{B}}$. For inference, no mask $\boldsymbol{M}^{\mathrm{N}}$ is used for TAS and anticipative mask $\boldsymbol{M}^{\mathrm{A}}$ for LTA.

Table 1: **Comparison with state of the art on TAS.** The overall results demonstrate the efficacy of ActFusion on TAS, achieving state-of-the-art performance across benchmark datasets. Bold values represent the highest accuracy, while underlined values indicate the second-highest accuracy.

| methods | 50 Salads [58] | | | | Breakfast [36] | | | | GTEA [21] | | | |
|---|---|---|---|---|---|---|---|---|---|---|---|---|
| | F1@{10, 25, 50} | edit | Acc. | Avg. | F1@{10, 25, 50} | edit | Acc. | Avg. | F1@{10, 25, 50} | Edit | Acc. | Avg. |
| MS-TCN [19] | 76.3 / 74.0 / 64.5 | 67.9 | 80.7 | 72.7 | 52.6 / 48.1 / 37.9 | 61.7 | 66.3 | 53.3 | 85.8 / 83.4 / 69.8 | 79.0 | 76.3 | 78.9 |
| MS-TCN++ [41] | 80.7 / 78.5 / 70.1 | 74.3 | 83.7 | 77.5 | 64.1 / 58.6 / 45.9 | 65.6 | 67.6 | 60.4 | 88.8 / 85.7 / 76.0 | 83.5 | 80.1 | 82.8 |
| SSTDA [12] | 83.0 / 81.5 / 73.8 | 75.8 | 83.2 | 79.5 | 75.0 / 69.1 / 55.2 | 73.7 | 70.2 | 68.6 | 90.0 / 89.1 / 78.0 | 86.2 | 79.8 | 84.6 |
| GTRM [30] | 75.4 / 72.8 / 63.9 | 67.5 | 82.6 | 72.4 | 57.5 / 54.0 / 43.3 | 58.7 | 65.0 | 55.7 | - / - / - | - | - | - |
| BCN [61] | 82.3 / 81.3 / 74.0 | 74.3 | 84.4 | 79.3 | 68.7 / 65.5 / 55.0 | 66.2 | 70.4 | 65.2 | 88.5 / 87.1 / 77.3 | 84.4 | 79.8 | 83.4 |
| MTDA [11] | 82.0 / 80.1 / 72.5 | 75.2 | 83.2 | 78.6 | 74.2 / 68.6 / 56.5 | 73.6 | 71.0 | 68.8 | 90.5 / 88.4 / 76.2 | 85.8 | 80.0 | 84.2 |
| Global2local [23] | 80.3 / 78.0 / 69.8 | 73.4 | 82.2 | 76.7 | 74.9 / 69.0 / 55.2 | 73.3 | 70.7 | 68.6 | 89.9 / 87.3 / 75.8 | 84.6 | 78.5 | 83.2 |
| HASR [3] | 86.6 / 85.7 / 78.5 | 81.0 | 83.9 | 83.1 | 74.7 / 69.5 / 57.0 | 71.9 | 69.4 | 68.5 | 90.9 / 88.6 / 76.4 | 87.5 | 78.7 | 84.4 |
| ASRF [32] | 84.9 / 83.5 / 77.3 | 79.3 | 84.5 | 81.9 | 74.3 / 68.9 / 56.1 | 72.4 | 67.6 | 67.9 | 89.4 / 87.8 / 79.8 | 83.7 | 77.3 | 83.6 |
| ASFormer [65] | 85.1 / 83.4 / 76.0 | 79.6 | 85.6 | 81.9 | 76.0 / 70.6 / 57.4 | 75.0 | 73.5 | 70.5 | 90.1 / 88.8 / 79.2 | 84.6 | 79.7 | 84.5 |
| ASFormer + KARI [24] | 85.4 / 83.8 / 77.4 | 79.9 | 85.3 | 82.4 | 78.8 / 73.7 / 60.8 | 77.8 | 74.0 | 73.0 | - / - / - | - | - | - |
| Temporal Agg. [52] | - / - / - | - | - | - | 59.2 / 53.9 / 39.5 | 54.5 | 64.5 | 54.3 | - / - / - | - | - | - |
| UARL [10] | 85.3 / 83.5 / 77.8 | 78.2 | 84.1 | 81.8 | 65.2 / 59.4 / 47.4 | 66.2 | 67.8 | 61.2 | 92.7 / 91.5 / 82.8 | 88.1 | 79.6 | 86.9 |
| DPRN [46] | 87.8 / 86.3 / 79.4 | 82.0 | 87.2 | 84.5 | 75.6 / 70.5 / 57.6 | 75.1 | 71.7 | 70.1 | 92.9 / 92.0 / 82.9 | 90.9 | 82.0 | 88.1 |
| SEDT [35] | 89.9 / 88.7 / 81.1 | 84.7 | 86.5 | 86.2 | - / - / - | - | - | - | 93.7 / 92.4 / 84.0 | 91.3 | 81.3 | 88.5 |
| TCTr [4] | 87.5 / 86.1 / 80.2 | 83.4 | 86.6 | 84.8 | 76.6 / 71.1 / 58.5 | 76.1 | 77.5 | 72.0 | 91.3 / 90.1 / 80.0 | 87.9 | 81.1 | 86.1 |
| FAMMSDTN [17] | 86.2 / 84.4 / 77.9 | 79.9 | 86.4 | 83.0 | 78.5 / 72.9 / 60.2 | 77.5 | 74.8 | 72.8 | 91.6 / 90.9 / 80.9 | 88.3 | 80.7 | 86.5 |
| DTL [63] | 87.1 / 85.7 / 78.5 | 80.5 | 86.9 | 83.7 | 78.8 / 74.5 / 62.9 | 77.7 | 75.8 | 73.9 | - / - / - | - | - | - |
| UVAST [8] | 89.1 / 87.6 / 81.7 | 83.9 | 87.4 | 85.9 | 76.9 / 71.5 / 58.0 | 77.1 | 69.7 | 70.6 | 92.7 / 91.3 / 81.0 | 92.1 | 80.2 | 87.5 |
| BrPrompt [40] | 89.2 / 87.8 / 81.3 | 83.8 | 88.1 | 86.0 | - / - / - | - | - | - | 94.1 / 92.0 / 83.0 | 91.6 | 81.2 | 88.4 |
| MCFM [31] | 90.6 / 89.5 / 84.2 | 84.6 | 90.3 | 87.8 | - / - / - | - | - | - | 91.8 / 91.2 / 80.8 | 88.0 | 80.5 | 86.5 |
| LTContext [5] | 89.4 / 87.7 / 82.0 | 83.2 | 87.7 | 86.0 | 77.6 / 72.6 / 60.1 | 77.0 | 74.2 | 72.3 | - / - / - | - | - | - |
| DiffAct [43] | 90.1 / 89.2 / 83.7 | 85.0 | 88.9 | 87.4 | 80.3 / 75.9 / 64.6 | 78.4 | 76.4 | 75.1 | 92.5 / 91.5 / 84.7 | 89.6 | 82.2 | 88.1 |
| **ActFusion (ours)** | **91.6 / 90.7 / 84.8** | **86.0** | 89.3 | **88.5** | **81.0 / 76.2 / 64.7** | **79.3** | 76.4 | **75.5** | **94.1 / 93.3 / 86.9** | 91.6 | 81.9 | **89.6** |

## 4.4 Training objective

The model is trained with three types of losses: cross-entropy loss $\mathcal{L}^{\mathrm{ce}}$ for frame-wise classification, boundary smoothing loss $\mathcal{L}^{\mathrm{smo}}$ [19], and boundary alignment loss $\mathcal{L}^{\mathrm{bd}}$ [43]. These losses are applied to both encoder and decoder, where the encoder serves as an auxiliary task to enhance discrimination ability. Given the ground-truth action label $\boldsymbol{A}^0 \in \mathbb{R}^{T \times K}$ and the predictions $\hat{\boldsymbol{A}} \in \mathbb{R}^{T \times K}$, the cross entropy loss is defined by:

$$\mathcal{L}_{\mathrm{ce}}(\boldsymbol{A}^0, \hat{\boldsymbol{A}}) = -\frac{1}{T} \sum_{i=1}^{T} \sum_{j=1}^{K} \boldsymbol{A}_{i,j}^0 \log \hat{\boldsymbol{A}}_{i,j}. \tag{8}$$

To prevent over-segmentation errors, a temporal smooth loss [19] between adjacent frames based on a truncated mean squared error over the frame-wise log probabilities are defined by:

$$\mathcal{L}_{\mathrm{smo}}(\hat{\boldsymbol{A}}) = \frac{1}{(T-1)K} \sum_{i=1}^{T-1} \sum_{j=1}^{K} (\log \hat{\boldsymbol{A}}_{i,j} - \log \hat{\boldsymbol{A}}_{i+1,j})^2, \tag{9}$$

where the difference of the log probabilities of two adjacent frames is truncated with a threshold value. For precise boundary detection, the boundary alignment loss [43] is employed based on the binary cross-entropy loss:

$$\mathcal{L}_{\mathrm{bd}}(\bar{\boldsymbol{B}}, \hat{\boldsymbol{A}}) = \frac{1}{T-1} \sum_{i=1}^{T-1} \{-\bar{\boldsymbol{B}}_i \log(1 - \hat{\boldsymbol{A}}_i \cdot \hat{\boldsymbol{A}}_{i+1}) - (1 - \bar{\boldsymbol{B}}_i) \log(\hat{\boldsymbol{A}}_i \cdot \hat{\boldsymbol{A}}_{i+1})\}, \tag{10}$$

where $\bar{\boldsymbol{B}} = \kappa(\boldsymbol{B})$ represents a softened version of the ground-truth action boundary $\boldsymbol{B}_i = \mathbb{1}(\boldsymbol{A}_i^0 \neq \boldsymbol{A}_{i+1}^0)$, achieved through a Gaussian kernel $\kappa$. The encoder and decoder losses are defined by

$$\mathcal{L}^{\mathrm{enc}} = \lambda_{\mathrm{ce}}^{\mathrm{enc}} \mathcal{L}_{\mathrm{ce}}^{\mathrm{enc}}(\boldsymbol{A}^0, \hat{\boldsymbol{A}}^{\mathrm{enc}}) + \lambda_{\mathrm{smo}}^{\mathrm{enc}} \mathcal{L}_{\mathrm{smo}}^{\mathrm{enc}}(\hat{\boldsymbol{A}}^{\mathrm{enc}}) + \lambda_{\mathrm{bd}}^{\mathrm{enc}} \mathcal{L}_{\mathrm{bd}}^{\mathrm{enc}}(\bar{\boldsymbol{B}}, \hat{\boldsymbol{A}}^{\mathrm{enc}}), \tag{11}$$

$$\mathcal{L}^{\mathrm{dec}} = \lambda_{\mathrm{ce}}^{\mathrm{dec}} \mathcal{L}_{\mathrm{ce}}^{\mathrm{dec}}(\boldsymbol{A}^0, \hat{\boldsymbol{A}}^{\mathrm{dec}}) + \lambda_{\mathrm{smo}}^{\mathrm{dec}} \mathcal{L}_{\mathrm{smo}}^{\mathrm{dec}}(\hat{\boldsymbol{A}}^{\mathrm{dec}}) + \lambda_{\mathrm{bd}}^{\mathrm{dec}} \mathcal{L}_{\mathrm{bd}}^{\mathrm{dec}}(\bar{\boldsymbol{B}}, \hat{\boldsymbol{A}}^{\mathrm{dec}}), \tag{12}$$

where $\lambda$ denotes a scaling factor. The total loss is defined as $\mathcal{L}^{\mathrm{total}} = \mathcal{L}^{\mathrm{enc}} + \mathcal{L}^{\mathrm{dec}}$.

## 5 Experiments

In this section, we conduct experiments to demonstrate the effectiveness of our model. All reported experimental results are obtained from inference using a single unified model. We evaluate our

Table 2: **Comparison with the state of the art on LTA.** The overall results demonstrate the effectiveness of ActFusion, achieving new SOTA performance in LTA. Bold values represent the highest accuracy, while underlined values indicate the second-highest accuracy.

| dataset | input type | methods | $\beta\ (\alpha=0.2)$ | | | | $\beta\ (\alpha=0.3)$ | | | |
|---|---|---|---|---|---|---|---|---|---|---|
| | | | 0.1 | 0.2 | 0.3 | 0.5 | 0.1 | 0.2 | 0.3 | 0.5 |
| 50 Salads | label | RNN [2] | 30.06 | 25.43 | 18.74 | 13.49 | 30.77 | 17.19 | 14.79 | 09.77 |
| | | CNN [2] | 21.24 | 19.03 | 15.98 | 09.87 | 29.14 | 20.14 | 17.46 | 10.86 |
| | | UAAA (mode) [1] | 24.86 | 22.37 | 19.88 | 12.82 | 29.10 | 20.50 | 15.28 | 12.31 |
| | | Time-Cond. [34] | 32.51 | 27.61 | 21.26 | 15.99 | 35.12 | 27.05 | 22.05 | 15.59 |
| | feature | Temporal Agg. [52] | 25.50 | 19.90 | 18.20 | 15.10 | 30.60 | 22.50 | 19.10 | 11.20 |
| | | Cycle Cons. [20] | 34.76 | 28.41 | 21.82 | 15.25 | 34.39 | 23.70 | 18.95 | 15.89 |
| | | A-ACT [26] | 35.40 | **29.60** | 22.50 | 16.10 | 35.70 | 25.30 | 20.10 | 16.30 |
| | | FUTR [25] | **39.55** | 27.54 | 23.31 | 17.77 | 35.15 | 24.86 | 24.22 | 15.26 |
| | | ObjectPrompt [66] | 37.40 | 28.90 | **24.20** | 18.10 | 28.00 | 24.00 | **24.30** | 19.30 |
| | | **ActFusion (ours)** | **39.55** | 28.60 | 23.61 | **19.90** | **42.80** | **27.11** | 23.48 | **22.07** |
| Breakfast | label | RNN [2] | 18.11 | 17.20 | 15.94 | 15.81 | 21.64 | 20.02 | 19.73 | 19.21 |
| | | CNN [2] | 17.90 | 16.35 | 15.37 | 14.54 | 22.44 | 20.12 | 19.69 | 18.76 |
| | | UAAA (mode) [1] | 16.71 | 15.40 | 14.47 | 14.20 | 20.73 | 18.27 | 18.42 | 16.86 |
| | | Time-Cond. [34] | 18.41 | 17.21 | 16.42 | 15.84 | 22.75 | 20.44 | 19.64 | 19.75 |
| | feature | Temporal Agg. [52] | 24.20 | 21.10 | 20.00 | 18.10 | 30.40 | 26.30 | 23.80 | 21.20 |
| | | Cycle Cons. [20] | 25.88 | 23.42 | 22.42 | 21.54 | 29.66 | 27.37 | 25.58 | 25.20 |
| | | A-ACT [26] | 26.70 | 24.30 | 23.20 | 21.70 | 30.80 | 28.30 | 26.10 | 25.80 |
| | | FUTR [25] | 27.70 | 24.55 | 22.83 | 22.04 | 32.27 | 29.88 | 27.49 | 25.87 |
| | | **ActFusion (ours)** | **28.25** | **25.52** | **24.66** | **23.25** | **35.79** | **31.76** | **29.64** | **28.78** |

method on three widely-used benchmark datasets: 50 Salads [58], Breakfast [36], and GTEA [21] (see Sec. F for details). All three datasets are used to evaluate on TAS, while 50 Salads and Breakfast are used for evaluating LTA, following the protocols of the previous work [19, 65, 43, 20, 52, 25].

**Evaluation metrics.** For evaluation metrics for TAS, we report F1@$\{10, 25, 50\}$ scores, the edit score, and frame-wise accuracy [19, 65, 43]. The F1 scores and the edit score are segment-wise metrics, and accuracy is a frame-wise metric. Mean over classes accuracy (MoC) is adopted as an evaluation metric for LTA [2, 34, 52, 25].

**Implementation details** We utilize the pre-trained I3D features [9] as input video features for all datasets provided by [19]. For the diffusion process, we set the entire time step $S$ as 1000 [29, 54], with a skipped time step for inference set to 25 [54]. For the anticipation mask $M^{\text{A}}$, we set the observation ratio $\alpha \in \{0.2, 0.3, 0.4, 0.5, 0.6, 0.7, 0.8\}$. For the random mask $M^{\text{R}}$, we fix the size of patch $w$ to 10 for both tasks and randomly select the number of masked patches $N^{\text{R}}$ to 25, 10, and 20 for 50 Salads, Breakfast, and GTEA, respectively. For the encoder and decoder, we adopt a modified version of ASFormer [65] using relative positional bias where the maximum number of neighbors is set to 100. During inference, we set $\alpha$ to 1 and $\beta$ to 0 for TAS [65, 19, 43]. For LTA, we set $\alpha \in \{0.2, 0.3\}$ and $\beta \in \{0.1, 0.2, 0.3, 0.5\}$, following the evaluation protocols [2, 20, 52, 25]. See Sec. G for more details.

## 5.1 Comparison with the state of the art on TAS and LTA

Tables 1 and 2 present performance comparisons across benchmark datasets, where our single unified model achieves superior results on both tasks. Table 1 shows the results on TAS, where ActFusion outperforms all task-specific models demonstrating the benefits of joint learning. Table 2 compares LTA performance across different datasets and input types: predicted action labels of the visual features [49] and pre-trained I3D features [9]. Overall, ActFusion achieves state-of-the-art performance on two benchmark datasets, demonstrating the efficacy of joint learning for TAS and LTA based on the diffusion process. Note that we do not include the performance of ANTICIPATR [44] due to differences in evaluation setup, as reported in [67].

## 5.2 Analysis

We conduct comprehensive analyses to validate the effectiveness of the proposed method. In the following experiments, we evaluate our approach on the 50 Salads dataset. All experimental settings are the same as explained in Sec. 5 unless otherwise specified.

Table 3: **Segmentation helps anticipation**

| | $\mathcal{L}^{\text{enc}}$ | $\mathcal{L}_{\text{O}}^{\text{dec}}$ | $\mathcal{L}_{\text{A}}^{\text{dec}}$ | $\beta$ ($\alpha = 0.2$) | | | | $\beta$ ($\alpha = 0.3$) | | | |
|---|---|---|---|---|---|---|---|---|---|---|---|
| | | | | 0.1 | 0.2 | 0.3 | 0.5 | 0.1 | 0.2 | 0.3 | 0.5 |
| (1) | - | - | ✓ | 27.32 | 21.97 | 18.04 | 17.47 | 17.78 | 14.82 | 13.23 | 14.24 |
| (2) | - | ✓ | ✓ | 34.51 | 26.26 | 19.69 | 15.84 | 28.75 | 20.33 | 20.03 | 16.95 |
| (3) | ✓ | ✓ | ✓ | **39.55** | **28.60** | **23.61** | **19.90** | **42.80** | **27.11** | **23.48** | **22.07** |

Table 4: **Ablation studies on masking**

(a) Results on TAS

| | $M^{\text{N}}$ | $M^{\text{A}}$ | $M^{\text{R}}$ | $M^{\text{B}}$ | $M^{\text{S}}$ | F1@{10, 25, 50} | Edit | Acc. | Avg. |
|---|---|---|---|---|---|---|---|---|---|
| (1) | ✓ | ✓ | ✓ | ✓ | ✓ | **91.6 / 90.7 / 84.8** | **86.0** | **89.3** | **88.5** |
| (2) | ✓ | - | ✓ | ✓ | ✓ | 90.0 / 88.9 / 83.9 | 84.7 | 88.6 | 87.2 |
| (3) | ✓ | ✓ | - | ✓ | ✓ | 89.9 / 89.0 / 82.4 | 83.9 | 88.7 | 86.8 |
| (4) | ✓ | ✓ | ✓ | - | ✓ | 89.9 / 89.3 / 83.9 | 83.7 | 88.3 | 87.0 |
| (5) | ✓ | ✓ | ✓ | ✓ | - | 90.2 / 89.2 / 83.7 | 85.0 | 87.7 | 87.2 |

(b) Results on LTA

| | $M^{\text{N}}$ | $M^{\text{A}}$ | $M^{\text{R}}$ | $M^{\text{B}}$ | $M^{\text{S}}$ | $\beta$ ($\alpha = 0.2$) | | | |
|---|---|---|---|---|---|---|---|---|---|
| | | | | | | 0.1 | 0.2 | 0.3 | 0.5 |
| (1) | ✓ | ✓ | ✓ | ✓ | ✓ | **42.80** | 27.11 | **23.48** | **22.07** |
| (2) | ✓ | - | ✓ | ✓ | ✓ | 35.24 | 23.15 | 15.72 | 9.38 |
| (3) | ✓ | ✓ | - | ✓ | ✓ | 38.60 | 25.57 | 20.42 | 18.62 |
| (4) | ✓ | ✓ | ✓ | - | ✓ | 37.23 | 25.29 | 20.82 | 20.35 |
| (5) | ✓ | ✓ | ✓ | ✓ | - | 41.42 | **27.38** | 22.54 | 19.32 |

**Segmentation helps anticipation.** To validate the impact of learning TAS on LTA, we conduct loss ablation experiments by removing the action segmentation loss on the observed $N^{\text{O}}$ frames. Table 3 shows the results. Specifically, we exclude the encoder loss $\mathcal{L}^{\text{enc}}$ in Eq. 11, while omitting the decoder loss $\mathcal{L}^{\text{dec}}$ in Eq. 12 on the observed frames. For simplicity, we denote the decoder loss for the observed frames and unobserved frames to be anticipated as $\mathcal{L}_{\text{O}}^{\text{dec}}$ and $\mathcal{L}_{\text{A}}^{\text{dec}}$, respectively. Note that $\mathcal{L}_{\text{O}}^{\text{dec}}$ are utilized for effective anticipation learning. By comparing (1) and (3) in Table 3, we observe a significant performance drop when removing the loss on the observed frames. We observe that applying segmentation loss on the observed frames in the decoder improves performance when comparing (1) and (3). The overall results show that learning action segmentation plays a crucial role in improving action anticipation.

**Anticipation helps segmentation.** To validate the effect of jointly learning anticipation along with segmentation, we conduct ablation studies on anticipative masking $M^{\text{A}}$ in Table 4. Comparing (1) and (2) in Table 4a, we find that using anticipation masking helps improve the performance of TAS. In comparison to the results in Table 4, where segmentation greatly enhances anticipation, the performance improvement without using anticipation masking is slightly lower. Nevertheless, we find that anticipation does contribute positively to the overall metrics of segmentation.

**Ablation studies on masking types**. To evaluate the effects of different types of masking on TAS and LTA, we conduct ablation studies in Table 4. The performance on TAS and LTA are presented in Table 4a and 4b, respectively. We observe that anticipative masking $M^{\text{A}}$ plays a significant role in the joint learning of TAS and LTA, as evidenced by the substantial performance drop in LTA when comparing row (2) with the other rows in Table 4b. Random masking $M^{\text{R}}$ significantly reduces performance in both TAS and LTA, as shown in row (3) of Table 4. Furthermore, the use of boundary masking $M^{\text{B}}$ and relation masking $M^{\text{S}}$ is also essential for both tasks. Due to the limited space, we only report the performance on LTA when the observation ratio $\alpha$ is set to 0.3.

**Effects of learnable mask tokens.** To verify the effects of learnable mask tokens $m$, we replace $m$ with zero vectors. Table 5 shows the overall results. Comparing rows (1) and (2) in Table 5, we find that using learnable mask tokens is more effective for both TAS and LTA. We conjecture that this is due to the mask tokens being embedded with the visual tokens within the encoder, which aids in effectively anticipating the invisible parts.

**Position of mask tokens.** Instead of providing mask tokens as input to the encoder, we conduct experiments on providing mask tokens as input to the decoder, as shown in row (3) in Table 5. In this experiment, only the visible frames are given as input to the encoder, while mask tokens $m' \in \mathbb{R}^{1 \times D}$ are applied in the decoder to fill the original masked positions, similar to the approach used in masked auto-encoder [28]. By comparing rows (2) and (3) in Table 5, we observe that using mask tokens in the encoder is more effective than using them in the decoder. We hypothesize that embedding mask tokens alongside visual tokens within the encoder benefits joint learning of TAS and LTA. Furthermore, when mask tokens are provided only to the decoder, they do not receive the encoder loss $\mathcal{L}^{\text{enc}}$, which may negatively impact performance.

Table 5: **Analysis on mask tokens**

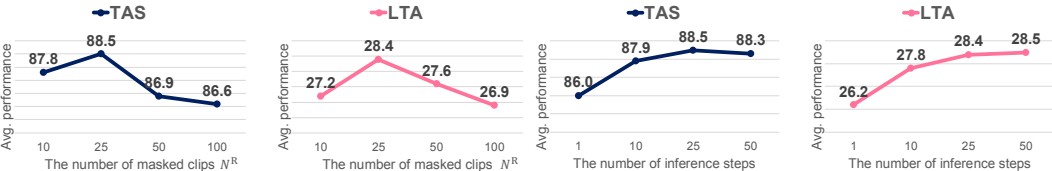

(a) Results on TAS

|  | Pos. | Learn. | F1@{10, 25, 50} | Edit | Acc. | Avg. |
|---|---|---|---|---|---|---|
| (1) | enc | - | 90.5 / 89.1 / 83.3 | 84.3 | 87.4 | 86.9 |
| (2) | enc | ✓ | **91.6 / 90.7 / 84.8** | **86.0** | **89.3** | **88.5** |
| (3) | dec | ✓ | 89.9 / 88.9 / 82.1 | 84.2 | 88.1 | 86.7 |

(b) Results on LTA

|  | Pos | Learn | $\beta\ (\alpha = 0.2)$ | | | | $\beta\ (\alpha = 0.3)$ | | | |
|---|---|---|---|---|---|---|---|---|---|---|
|  |  |  | 0.1 | 0.2 | 0.3 | 0.5 | 0.1 | 0.2 | 0.3 | 0.5 |
| (1) | enc | - | 35.89 | 27.51 | 21.48 | **20.70** | 38.03 | 26.91 | 22.69 | 21.88 |
| (2) | enc | ✓ | **39.55** | **28.60** | **23.61** | 19.90 | **42.80** | **27.11** | **23.48** | **22.07** |
| (3) | dec | ✓ | 33.99 | 23.61 | 18.50 | 12.13 | 38.25 | 23.46 | 17.87 | 14.05 |

Figure 3: **The number of masked clips** $N^{\mathrm{R}}$

Figure 4: **Inference steps of diffusion process**

**The number of masked clips $N^{\mathrm{R}}$ in random masking.** To determine the number of masked clips $N^{\mathrm{R}}$ in random masking, we conduct experiments by adjusting $N^{\mathrm{R}}$ from 10 to 100 while fixing the window size $Q$ of each masked clip $N^{\mathrm{P}}$ to 10. Figure 3 illustrates the result, suggesting that employing an appropriate amount of masking is crucial.

**The number of inference steps in the diffusion process.** We compare performance according to the number of inference steps of the diffusion process in Fig. 4. We observe consistent performance increases as the number of inference steps increases. As the increasing step requires more computation, we choose to use 25 steps for inference in all of our experiments.

## 5.3 Qualitative results

Figure 5 presents qualitative results evaluated on both TAS and LTA using a single model. Each figure includes video frames, ground-truth action sequences, and predicted results for TAS and LTA. Only the visible parts (observed frames) are used as input during inference on LTA. Overall results show that ActFusion effectively handles both visible and future segments, accurately classifying current actions and anticipating future ones. Additional results are provided in Fig. S3 and S4.

## 5.4 Evaluation without using ground truth prediction length on LTA

Benchmark evaluations in previous work, including those of [20, 25] and our results presented above, have been typically conducted following the evaluation protocol of [2] where prediction length $N^{\mathrm{A}}$ is set to $\beta T$ in testing; $\beta$ and $T$ are the prediction ratio and the ground-truth video length, respectively. This can be seen as a leakage of ground-truth information in testing because the exact length of future actions is supposed to be inherently unknown during inference in real-world scenarios. In this subsection, we thus rectify the evaluation setting by determining the prediction length as $rN^{\mathrm{O}}$ based solely on the number of observed frames $N^{\mathrm{O}}$, where $r$ is a hyperparameter that adjusts the relative length of future predictions. We then train our method with this modified anticipation masking strategy, denoted as ActFusion†. In this experiment, $r$ is set to 4.

Using the rectified evaluation protocol, we compare ours with Cycle Cons. [20] and FUTR [25], whose codes are available[1]. For a fair comparison, we modify both models to use the same prediction length $rN^{\mathrm{O}}$, denoted as Cycle Cons.† and FUTR†. The results are summarized in Table 6 where the numbers in parentheses indicate relative performance changes compared to those in Table 2. Table 6 shows that all the methods exhibit overall performance degradation when ground-truth length information is not used. These observations reveal that previous results have benefited from the use of ground-truth length $T$, leading to information leakage in testing. Therefore, we suggest that future research avoid relying on this information for a more realistic and fair comparison. On the other hand, the results also show that our approach consistently outperforms the methods across benchmark datasets, demonstrating its effectiveness even without ground-truth prediction length information. Please refer to Sec. G for further details of the experimental setup.

---

[1]The official code for Cycle Cons. [20] was obtained directly from the authors, and the official code repository for FUTR [25] is available at https://github.com/gongda0e/FUTR.

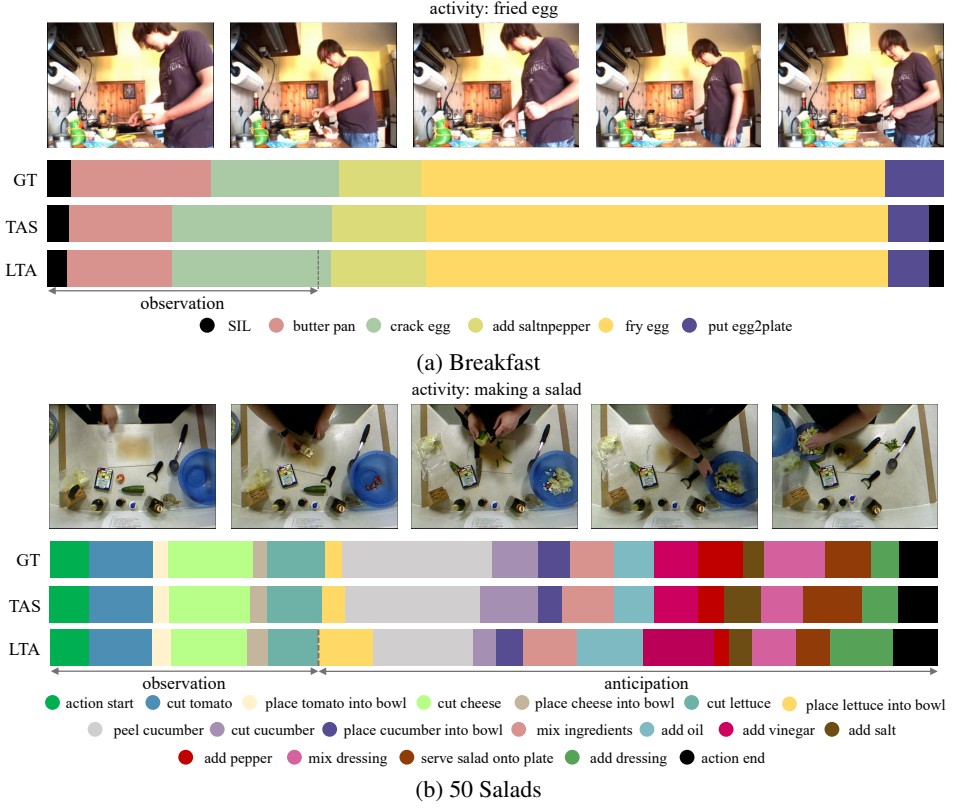

Figure 5: **Qualitative results**

Table 6: **Evaluation without using ground truth prediction length on LTA**

| dataset | methods | $\beta$ ($\alpha = 0.2$) | | | | $\beta$ ($\alpha = 0.3$) | | | |
|---|---|---|---|---|---|---|---|---|---|
| | | 0.1 | 0.2 | 0.3 | 0.5 | 0.1 | 0.2 | 0.3 | 0.5 |
| 50 Salads | Cycle Cons.[†] [20] | 31.85 (2.91↓) | 28.19 (0.22↓) | 23.98 (2.16↓) | 16.02 (0.77↓) | 26.54 (7.85↓) | 18.36 (5.34↓) | 14.52 (4.43↓) | 10.34 (5.55↓) |
| | FUTR[†] [25] | 32.29 (4.71↓) | 24.49 (3.33↓) | 20.00 (2.46↓) | 13.89 (2.87↓) | 21.52 (11.83↓) | 15.74 (9.44↓) | 11.89 (8.25↓) | 7.48 (8.04↓) |
| | **ActFusion (ours)**[†] | **35.86** (3.69↓) | **28.09** (0.51↓) | **24.20** (0.59↑) | **20.13** (0.23↑) | **41.14** (1.66↓) | **25.68** (1.43↓) | **23.12** (0.36↓) | **18.35** (3.71↓) |
| Breakfast | Cycle Cons.[†] [20] | 25.65 (0.23↓) | 23.10 (0.32↓) | 21.66 (0.76↓) | 19.73 (1.82↓) | 29.08 (0.59↓) | 25.68 (1.7↓) | 23.00 (2.59↓) | 20.84 (4.36↓) |
| | FUTR[†] [25] | 27.85 (0.16↑) | 24.68 (0.15↑) | 22.91 (0.09↑) | 20.98 (1.06↓) | 32.36 (0.09↑) | 28.96 (0.92↓) | 25.49 (2.02↓) | 23.56 (2.33↓) |
| | **ActFusion (ours)**[†] | **27.90** (0.70↓) | **24.69** (0.82↓) | **22.99** (2.58↓) | **22.42** (0.84↓) | **33.47** (1.75↓) | **30.28** (1.22↓) | **30.69** (1.12↑) | **26.77** (2.02↓) |

# 6 Conclusion

We have presented ActFusion, a unified diffusion model that jointly addresses temporal action segmentation and long-term action anticipation in videos. The key to our method is anticipative masking, where learnable mask tokens replace unobserved frames, enabling simultaneous action segmentation of visible parts and anticipation of invisible parts. Our comprehensive experiments demonstrate that this unified approach not only outperforms existing task-specific models on both tasks but also reveals the mutual benefits of joint learning. Additionally, by evaluating our method both with and without ground-truth length information during LTA inference, we hope to motivate future research toward not using this information during testing. We believe that ActFusion demonstrates the potential of unifying complementary tasks in temporal action understanding, opening new directions for future research.

# 7 Acknowledgement

This work was supported by Samsung Electronics (IO201208-07822-01) and IITP grants (RS-2022-II220959: Few-Shot Learning of Causal Inference in Vision and Language for Decision Making (50%), RS-2022-II220264: Comprehensive Video Understanding and Generation with Knowledge-based Deep Logic Neural Network (45%), RS-2019-II191906: AI Graduate School Program at POSTECH (5%)) funded by Ministry of Science and ICT, Korea. We thank reviewers for pointing out the issue of the evaluation protocol and providing insightful discussions.

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

## Appendix

In this appendix, we offer detailed descriptions and additional results, which are omitted in the main paper due to the lack of space. We provide a detailed explanation of diffusion models in Sec. A, training and inference algorithms in Sec. B, masking types in Sec. C, encoder and decoder layers in Sec. D, additional experimental results in Sec. E, more information of datasets in Sec. F, more experimental details in Sec. G, more qualitative results in Sec. H, and discussions of limitations and broader impacts in Sec. I.

## A  Diffusion models

In this section, we give an in-depth explanation of the diffusion models [29, 54] described in Sec. 3. The forward process adds noise to a data distribution $x^0 \sim q(x^0)$, following the noise distribution $q$ based on the Markov property. Specifically, the forward process is defined by adding noise at an arbitrary time step $s$ with a variance $\gamma(s)$ as follows:

$$q(x^1, ..., x^S | x^0) := \prod_{s=1}^{S} q(x^s | x^{s-1}), \tag{13}$$

$$q(x^s | x^{s-1}) := N(x^s; \sqrt{1 - \gamma(s)} x^{s-1}, \gamma(s)\mathbf{I}), \tag{14}$$

where $S$ represents the entire time step. We can directly obtain the noisy data distribution $x^s$ at time step $s$ without iteratively applying $q$ due to the Markov property:

$$q(x^s | x^0) := N(x^s; \sqrt{\bar{\delta}(s)} x^0, (1 - \bar{\delta}(s))\mathbf{I}), \tag{15}$$

$$x^s := \sqrt{\bar{\delta}(s)} x^0 + \sqrt{1 - \bar{\delta}(s)} \epsilon, \tag{16}$$

where $\delta(s) := 1 - \gamma(s)$, $\bar{\delta}(s) := \prod_{i=1}^{s} \delta(i)$, and $\epsilon \sim N(0, \mathbf{I})$. Instead of using $\gamma(s)$, $1 - \delta(s)$ is utilized [29]. The posterior $q(x^{s-1} | x^s, x^0)$ conforms to a Gaussian distribution and is expressed in terms of mean $\tilde{m}^s(x^s, x^0)$ and variance $\tilde{\gamma}(s)$ using Bayes Theorem:

$$q(x^{s-1} | x^s, x^0) = N(x^{s-1}; \tilde{m}(x^s, x^0), \tilde{\gamma}(s)\mathbf{I}), \tag{17}$$

where

$$\tilde{m}^s(x^s, x^0) := \frac{\sqrt{\bar{\delta}(s-1)} \gamma(s)}{1 - \bar{\delta}(s)} x^0 + \frac{\sqrt{\delta(s)}(1 - \bar{\delta}(s-1))}{1 - \bar{\delta}(s)} x^s, \tag{18}$$

$$\tilde{\gamma}(s) := \frac{1 - \bar{\delta}(s-1)}{1 - \bar{\delta}(s)} \gamma(s). \tag{19}$$

With a sufficiently large $S$ and an appropriate variance schedule $\gamma(s)$, the noisy data $x^S$ follows an isotropic Gaussian distribution. Consequently, if we know the reverse distribution $q(x^{s-1} | x^s)$, we can sample $x^S \sim N(0, \mathbf{I})$ and reverse the process to obtain a sample from $q(x^0)$. However, since $q(x^{s-1} | x^s)$ relies on the entire data distribution, we employ a neural network to approximate $q(x^{s-1} | x^s)$ as follows:

$$p_\theta(x^{s-1} | x^s) := N(x^{s-1}; m_\theta(x^s, s), \Sigma_\theta(x^s, s)), \tag{20}$$

where $m_\theta$ and $\Sigma_\theta$ represent the predicted mean and co-variance, respectively, derived from the neural network. Predicting $\epsilon$ or $x^0$ in Eq. 16 is found to be effective [29]. In this work, we choose to predict $x^0$ using the neural network.

**Algorithm 1** ActFusion Training

```python
def train(f, a_gt):
    """
    f: video features [B, T, C]
    a_gt: ground-truth action labels [B, T, K]
    # B: batch
    # T: number of frames
    # C: video feature dimension
    # K: number of action classes
    """
    # masking input features
    mask_types = ['none', 'ant', 'random', 'boundary', 'rel']
    mask_type = random.choice(mask_types)
    f_prime = mask(f, mask_type)

    # video embeddings and predictions from encoder
    e, a_hat_enc = encoder(f_prime)

    # signal scaling
    a_gt = (a_gt * 2 - 1) * scale

    # corrupt a_gt
    s = randint(0, S)           # time step
    eps = normal(mean=0, std=1) # noise: [B, T, K]
    a_crpt = sqrt(delta_cumprod(s)) * a_gt + sqrt(1 - delta_cumprod(s)) * eps

    # predictions from decoder
    a_hat_dec = decoder(a_crpt, e, s)

    # training loss
    loss_enc = cal_enc_loss(a_hat_enc, a_gt)
    loss_dec = cal_dec_loss(a_hat_dec, a_gt)
    loss_total = loss_enc + loss_dec

    return loss_total
```

# B   Algorithms

We provide training algorithms of ActFusion in Alg. 1 and inference algorithms for TAS and LTA in Alg. 2 and Alg. 3, respectively.

**Algorithm 2** ActFusion TAS Inference

```python
def inference(f, steps, S):
    """
    f: [B, T, C]
    steps: number of inference steps
    S: number of time steps
    """

    # masking
    f_prime = mask(f, 'none')

    # Encode video features
    e = encoder(f_prime)

    # sample noisy action label
    a_s = normal(mean=0, std=1)

    # uniform sampling step size
    times = reversed(linspace(-1, S, steps))

    # [(S-1, S-2), (S-2, S-3), ..., (1,0), (0,-1)]
    time_pairs = list(zip(times[:-1],times[1:]))
    for t_now, t_next in zip(time_pairs):
        # predict a_0 from a_s
        a_hat = decoder(a_s, e, t_now)

        # estimate a_s at t_next
        a_s = ddim_step(a_s, a_hat, t_now, t_next)

    return a_hat
```

delta_cumprod(s): cumulative product of $\delta_i$, , $\prod_{i=1}^{s} \delta_i$

**Algorithm 3** ActFusion LTA Inference

```python
def inference(f, steps, S):
    """
    f: [B, alpha*T, C]
    steps: number of inference steps
    S: number of time steps
    """

    # masking
    f_prime = mask(f, 'ant')

    # Encode video features
    e = encoder(f_prime)

    # sample noisy action label
    a_s = normal(mean=0, std=1)

    # uniform sampling step size
    times = reversed(linspace(-1, S, steps))

    # [(S-1, S-2), (S-2, S-3), ..., (1,0), (0,-1)]
    time_pairs = list(zip(times[:-1],times[1:]))
    for t_now, t_next in zip(time_pairs):
        # predict a_0 from a_s
        a_hat = decoder(a_s, e, t_now)

        # estimate a_s at t_next
        a_s = ddim_step(a_s, a_hat, t_now, t_next)

    return a_hat
```

## C  Masking types

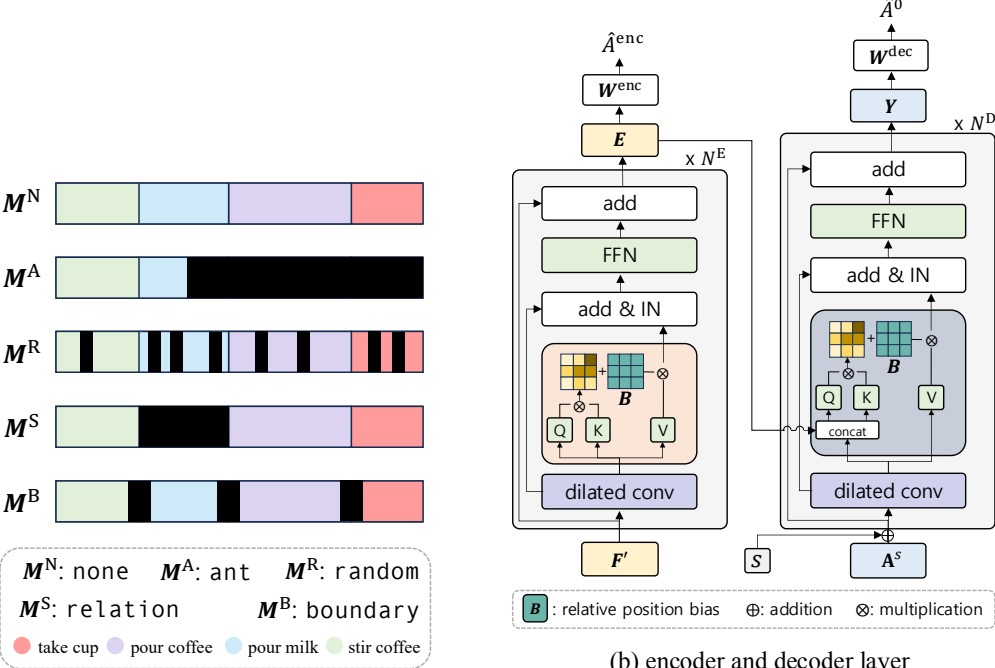

| | |
|---|---|
| Figure S1: **Masking types** | Figure S2: **Encoder & decoder layer** |

Figure S1 illustrates five types of masking. In Sec. 5.3, we introduced two types of masking: anticipative masking and random masking. The anticipation mask $M^{\mathrm{A}}$ and the random mask $M^{\mathrm{R}}$ are defined for each masking, respectively. Additionally, we adopt the relation mask $M^{\mathrm{S}}$ and the boundary mask $M^{\mathrm{B}}$ introduced in DiffAct [43]. The relation mask $M^{\mathrm{S}}$ is proposed to help the model learn relational dependencies between actions. It is defined by $M^{\mathrm{S}} = \mathbb{1}(A^0_{i,a} \neq 1), i \in \{1, 2, ..., T\}$ and $a \in \{1, 2, ..., K\}$, with $T$ representing the number of frames and $K$ denoting the number of action classes. Here, a specific action class $a$ is randomly selected for the mask. The boundary mask $M^{\mathrm{B}}$ hides visual features near the boundaries based on the soft ground truth $\bar{B}$ to manage the uncertainty associated with action transitions. Specifically, it is defined by $M^{\mathrm{B}}_i = \mathbb{1}(\bar{B} < 0.5)$, where $i \in \{1, 2, ..., T\}$.

During training, a masking type is randomly selected among five: no mask $M^{\mathrm{N}}$, anticipative mask $M^{\mathrm{A}}$, random mask $M^{\mathrm{R}}$, relation mask $M^{\mathrm{S}}$, and boundary mask $M^{\mathrm{B}}$, for each iteration as detailed in Sec. 4.4. During inference, the no mask $M^{\mathrm{N}}$ and the anticipative mask $M^{\mathrm{A}}$ are utilized following the evaluation protocol [19, 65, 43, 20, 52, 25].

## D  Model architecture

We employ a modified version of ASFormer [65] utilized in DiffAct [43] as the baseline model. An encoder $g$ consists of $N^{\mathrm{E}}$ number of layers and a decoder $h$ consists of $N^{\mathrm{D}}$ number of layers as described in Sec.4.2. Figure S2 illustrates detailed operations of the encoder and the decoder layers. An encoder layer consists of dilated convolution followed by dilated attention, instance normalization, and a feed-forward network. Additionally, we add relative position bias $B$ to attention scores to consider relative position relations among actions [48]. Similarly, a decoder layer comprises dilated convolution followed by dilated attention, instance normalization, and a feed-forward network. The output embedding $E$ from the encoder is concatenated to the input of the decoder after dilated convolutions. Subsequently, they are embedded as queries and keys in the dilated attention operation. We refer the reader to [65, 43] for more details.

Table S1: **Conditioning features**

(a) Results on TAS

| conditioning features | F1@{10, 25, 50} | Edit | Acc. | Avg. |
|:---:|:---:|:---:|:---:|:---:|
| $F'$ | 85.6 / 83.6 / 74.8 | 79.8 | 82.8 | 81.3 |
| $E$ | **91.6 / 90.7 / 84.8** | **86.0** | **89.3** | **88.5** |
| $\hat{A}$ | 90.5 / 89.4 / 83.2 | 84.6 | 87.6 | 87.1 |

(b) Results on LTA

| conditioning features | $\beta\,(\alpha = 0.2)$ | | | | $\beta\,(\alpha = 0.3)$ | | | |
|:---:|:---:|:---:|:---:|:---:|:---:|:---:|:---:|:---:|
| | 0.1 | 0.2 | 0.3 | 0.5 | 0.1 | 0.2 | 0.3 | 0.5 |
| $F'$ | 29.71 | 25.43 | 20.65 | 14.24 | 37.22 | 22.22 | 17.88 | 16.48 |
| $E$ | **39.55** | 28.60 | 23.61 | 19.90 | **42.80** | **27.11** | **23.48** | **22.07** |
| $\hat{A}$ | 39.45 | **29.33** | **24.23** | 19.98 | 37.96 | 25.41 | 22.76 | 21.18 |

Table S2: **Position embedding**

(a) Results on TAS

| position embedding | F1@{10, 25, 50} | Edit | Acc. | Avg. |
|:---:|:---:|:---:|:---:|:---:|
| none | 90.0 / 89.7 / **84.3** | 85.3 | 88.5 | 87.7 |
| Rel. bias | **91.6 / 90.7 / 84.8** | **86.0** | **89.3** | **88.5** |
| Rel. Emb. | 90.5 / 89.6 / 83.9 | 84.7 | 88.3 | 87.4 |
| Abs. Emb. | 87.4 / 86.2 / 78.7 | 80.5 | 83.9 | 83.3 |

(b) Results on LTA

| position embedding | $\beta\,(\alpha = 0.2)$ | | | | $\beta\,(\alpha = 0.3)$ | | | |
|:---:|:---:|:---:|:---:|:---:|:---:|:---:|:---:|:---:|
| | 0.1 | 0.2 | 0.3 | 0.5 | 0.1 | 0.2 | 0.3 | 0.5 |
| none | 35.33 | 26.41 | 22.74 | 17.89 | 41.52 | **27.60** | 22.53 | 18.69 |
| Rel. bias | **39.55** | **28.60** | **23.61** | **19.90** | **42.80** | 27.11 | **23.48** | **22.07** |
| Rel. Emb. | 37.37 | 27.32 | 22.44 | 18.50 | 42.47 | 26.62 | 22.51 | 20.78 |
| Abs. Emb. | 35.91 | 26.84 | 22.48 | 17.70 | 30.02 | 20.87 | 18.24 | 17.71 |

# E    Additional experiments

We provide additional experimental results, maintaining the same experimental settings as described in Sec.5.2 unless otherwise specified. The evaluation is conducted on the 50 Salads dataset.

**Conditioning features.** We investigate different types of conditioning features: masked features $F'$, output embeddings $E$ from the encoder, and the encoder prediction $\hat{A}$. Table S1 shows overall results, where using $E$ is more effective than using other features. The effectiveness of utilizing the encoder in our approach becomes apparent when comparing the first and second rows, where a significant performance drop is observed in the absence of the encoder. Since mask tokens replace visual tokens before going into the encoder, it is crucial for them to learn action relations through the encoder. Comparing the second and third rows, we also observe that utilizing features from intermediate layers yields slightly better results than using encoder predictions for both TAS and LTA.

**Position embedding.** In Table S2, we explore different types of position embeddings: relative position bias [48], relative position embedding [53], and absolute position embedding [15]. Comparing the first and second rows, we find that employing relative position bias enhances the overall performance in TAS and LTA. From the second and third rows, we observe that relative position bias is also more effective than using learnable relative position embedding. Using absolute position embedding leads to decreased performance in both TAS and LTA. We find that learnable embeddings often cause overfitting problems during training. As a result, we adopt relative position bias in our model.

Table S3: **Loss ablations**

(a) Results on TAS

| $L_{\text{bd}}$ | $L_{\text{smo}}$ | $L_{\text{ce}}$ | F1@10 | F1@25 | F1@50 | Edit | Acc. | Avg. |
|---|---|---|---|---|---|---|---|---|
| ✓ | | | 88.4 | 86.5 | 79.1 | 82.5 | 84.9 | 84.3 |
| ✓ | ✓ | | 91.3 | 90.0 | 84.5 | **86.3** | 88.8 | 88.2 |
| ✓ | ✓ | ✓ | **91.6** | **90.7** | **84.8** | 86.0 | **89.3** | **88.5** |

(b) Results on LTA

| $\mathcal{L}_{\text{bd}}$ | $\mathcal{L}_{\text{smo}}$ | $\mathcal{L}_{\text{ce}}$ | $\beta\ (\alpha = 0.2)$ | | | | $\beta\ (\alpha = 0.3)$ | | | |
|---|---|---|---|---|---|---|---|---|---|---|
| | | | 0.1 | 0.2 | 0.3 | 0.5 | 0.1 | 0.2 | 0.3 | 0.5 |
| ✓ | | | 35.62 | 27.04 | 20.17 | 15.93 | 34.38 | 22.33 | 19.96 | 16.94 |
| ✓ | ✓ | | 39.19 | **28.99** | 23.13 | 19.45 | 39.53 | 25.19 | 22.67 | 19.88 |
| ✓ | ✓ | ✓ | **39.55** | 28.60 | **23.61** | **19.90** | **42.80** | **27.11** | **23.48** | **22.07** |

Table S4: **Effects of reconstruction loss $\mathcal{L}_{\text{recon}}$**

(a) Results on TAS

| $\mathcal{L}_{\text{recon}}$ | F1@{10, 25, 50} | Edit | Acc. | Avg. |
|---|---|---|---|---|
| - | 85.6 / 83.6 / 74.8 | 79.8 | 82.8 | 81.3 |
| ✓ | **91.6 / 90.7 / 84.8** | **86.0** | **89.3** | **88.5** |

(b) Results on LTA

| $\mathcal{L}_{\text{recon}}$ | $\beta\ (\alpha = 0.2)$ | | | | $\beta\ (\alpha = 0.3)$ | | | |
|---|---|---|---|---|---|---|---|---|
| | 0.1 | 0.2 | 0.3 | 0.5 | 0.1 | 0.2 | 0.3 | 0.5 |
| - | 39.55 | 28.60 | 23.61 | **19.90** | 42.80 | **27.11** | **23.48** | **22.07** |
| ✓ | **40.80** | **31.02** | **25.59** | 13.94 | **46.56** | 26.22 | 18.56 | 16.15 |

**Loss ablations.** We conduct ablation studies on the loss functions: boundary loss, smoothing loss, and cross-entropy loss. Table R4 presents the results, demonstrating that the combination of bounding loss and smoothing loss is effective for both TAS and LTA. While the effectiveness of these losses in TAS is well-documented in previous research [19, 43, 65], their impact on LTA has been less explored. Notably, the smoothing loss leads to significant performance gains in both tasks, indicating that smoothed predictions are beneficial.

**Effects of reconstruction loss $\mathcal{L}_{\text{recon}}$** Masked auto-encoding is a technique used in training NLP models like BERT [15] and has recently been adapted to vision models [22, 28]. Inspired by this approach, we train our model to reconstruct input video features from the masked tokens as an auxiliary task. Specifically, we employ MLP layers on the encoder embeddings to reconstruct the input video features and apply mean squared error (MSE) loss $\mathcal{L}_{\text{recon}}$ during training.

Table S4 shows the overall results on both TAS and LTA tasks. In TAS, overall performance increases. We conjecture that reconstruction helps the model gain a deeper understanding of the underlying data structure and temporal dynamics by predicting the missing parts of the input. In LTA, we find that reconstruction is more effective on relatively short-term anticipation. Since short-term predictions are often based on more immediate context, there is less uncertainty. As a result, reconstructing masked features helps the model capture immediate patterns and transitions more accurately. However, for long-term predictions, as the model attempts to predict further into the future, the uncertainty increases significantly. Long-term predictions involve more variables and potential changes, making them inherently less predictable. This increased uncertainty might cause performance degradation, making reconstruction less effective for action anticipation.

Table S5: **Hyperparameters.** We provide the hyperparameters used during training for each dataset.

| hyperparameters | 50 Salads | Breakfast | GTEA |
|---|---|---|---|
| # of epochs | 5000 | 1000 | 10000 |
| batch size | 4 | 4 | 4 |
| sample rate | 8 | 1 | 1 |
| optimizer | Adam | Adam | Adam |
| learning rate | 0.0005 | 0.0001 | 0.0005 |
| weight decay | 0 | 0 | 1e$-$06 |
| # of encoder layers | 10 | 12 | 10 |
| # of decoder layers | 8 | 8 | 8 |
| dimension in encoder | 64 | 256 | 64 |
| dimension in decoder | 24 | 128 | 24 |
| conditioning features of $E$ layers | $\{5, 7, 9\}$ | $\{5, 7, 9\}$ | $\{7, 9\}$ |
| $\lambda_{ce}^{enc}$ | 0.5 | 0.5 | 0.5 |
| $\lambda_{smo}^{enc}$ | 0.1 | 0.025 | 0.1 |
| $\lambda_{bd}^{dec}$ | 0.0 | 0.0 | 0.0 |
| $\lambda_{ce}^{dec}$ | 0.5 | 0.5 | 0.5 |
| $\lambda_{smo}^{dec}$ | 0.1 | 0.025 | 0.1 |
| $\lambda_{bd}^{dec}$ | 0.1 | 0.1 | 0.1 |

# F   Datasets

The 50 Salads [58] dataset consists of 50 videos depicting 25 individuals preparing a salad. With over 4 hours of RGB-D video data, the annotations include 17 fine-grained action labels and 3 high-level activities. Notably, 50 Salads videos are longer than those in the Breakfast dataset, averaging 20 actions per video. The dataset is partitioned into 5 splits for cross-validations, and we report the average performance across all splits. The Breakfast dataset is under the license of Creative Commons Attribution-NonCommercial-ShareAlike 4.0 International License.

The Breakfast [36] dataset consists of 1,712 videos featuring 52 individuals preparing breakfast in 18 different kitchens. Each video is assigned to one of the 10 activities associated with breakfast preparation, utilizing 48 fine-grained action labels to define these activities. The average video duration is approximately 2.3 minutes, encompassing around 6 actions per video. The dataset is divided into 4 splits for cross-validations, and we report the average performance across all splits. The Breakfast dataset is under the license of Creative Commons Attribution 4.0 International License.

The GTEA dataset [21] comprises 28 videos capturing 11 action classes related to cooking activities. Each video includes 20 actions on average, and an average video duration is about a minute. The dataset is divided into 4 splits for cross-validations, and we report average performance across all splits. The GTEA dataset is under the license of Creative Commons Attribution 4.0 International License.

# G   Experimental details

**Cross-task generalization.** In Fig.1-(c), we conduct experiments on cross-task evaluations with DiffAct [43], FUTR [25], and TempAgg [52]. To evaluate DiffAct on LTA, we limit the input video frames to $F_{1:\alpha T}$, with zero masks appended to future frame lengths concatenated to the encoder output embeddings, adhering to DiffAct's masking strategies. Subsequently, the decoder predicts future actions based on the observed video frames. To evaluate FUTR on TAS, we utilize the encoder of FUTR as an action segmentation model. Note that we report the performance of TempAgg on both TAS and LTA as provided in the original paper [52].

**Evaluation without using ground-truth prediction length on LTA.** Baseline models of Cycle Cons. [20] and FUTR [25] anticipate future actions and their corresponding durations. The predicted durations are then directly multiplied by the ground-truth prediction length, $\beta T$, to generate the final predictions. In Table 6, we also experimented with using a modified prediction length, $r\alpha T$, instead

of the ground-truth length. Please note that we use a reproduced model for Cycle Cons. and the official model checkpoints for FUTR[2].

**Implementation details.** We provide additional implementation details to complement those described in Sec. 5.2. Table S5 presents the specific hyperparameters used in our experiments for each dataset. All experiments are conducted on a single NVIDIA RTX-3080 GPU. We implement ActFusion using Pytorch [47] and some of the official code repository of DiffAct [43][3] licensed under an MIT License.

# H   Qualitative results

We provide additional qualitative results for both successful and failure cases in Fig. S3 and Fig. S4, respectively. Figure S3 demonstrates the promising results on both TAS and LTA, showing the efficacy of joint learning these two tasks. In the failure cases, figure S4a highlights the importance of accurate segmentation on anticipation, where inaccurate action segmentation of the observed frames leads to wrong future anticipation. In Fig. S4b, ActFusion fails to detect the 'take knife' action class in action segmentation. However, the model anticipates missing action classes that are not observed in the observed frames. This implies that the model can infer missing actions based on the action relations of the observed and unobserved actions.

# I   Discussion

In this paper, we have proposed a unified diffusion model for action segmentation and anticipation, dubbed ActFusion. We have demonstrated the effectiveness of the proposed method through comprehensive experimental results, showing the bi-directional benefits of joint learning the two tasks. In this section, we will discuss the limitations, future work, and broader impact of the proposed method.

**Limitations and future work.** The proposed method shows the state-of-the-art results on TAS across three benchmark datasets. However, the frame-wise accuracy is slightly below compared to DiffAct [43]. We hypothesize that this discrepancy arises from the different masking strategies employed. In DiffAct, masking is applied to the output embeddings of the encoder, ensuring that the encoder always fully observes the visual features from the entire video. However, in our approach, masking is applied before the encoders, allowing for handling both visible and invisible tokens to unify the two tasks effectively. Consequently, the encoder may not always observe the entire visual features, potentially leading to slightly lower accuracy. This issue could potentially be addressed by employing reconstruction methods for masked features, similar to masked autoencoders [28, 22]. By further training the model to reconstruct the original features from the masked features, the encoder could be empowered to handle the masked features better. We leave this as our future work.

In future work, additional activity information could be integrated, particularly focusing on segment-wise action relations [8]. Moreover, exploring weakly-supervised learning approaches [39, 49, 37] could be beneficial for further enhancing the capabilities of our method.

**Broader impact.** To the best of our knowledge, we are the first to integrate action segmentation and anticipation within a unified model. We believe that our work lays the foundation for integrating these two tasks, offering substantial potential for real-world applications such as robots.

---

[2]Official model checkpoints of FUTR [25] are available at `https://github.com/gongda0e/FUTR`.
[3]Official code repository of DiffAct [43] is available at `https://github.com/Finspire13/DiffAct`.

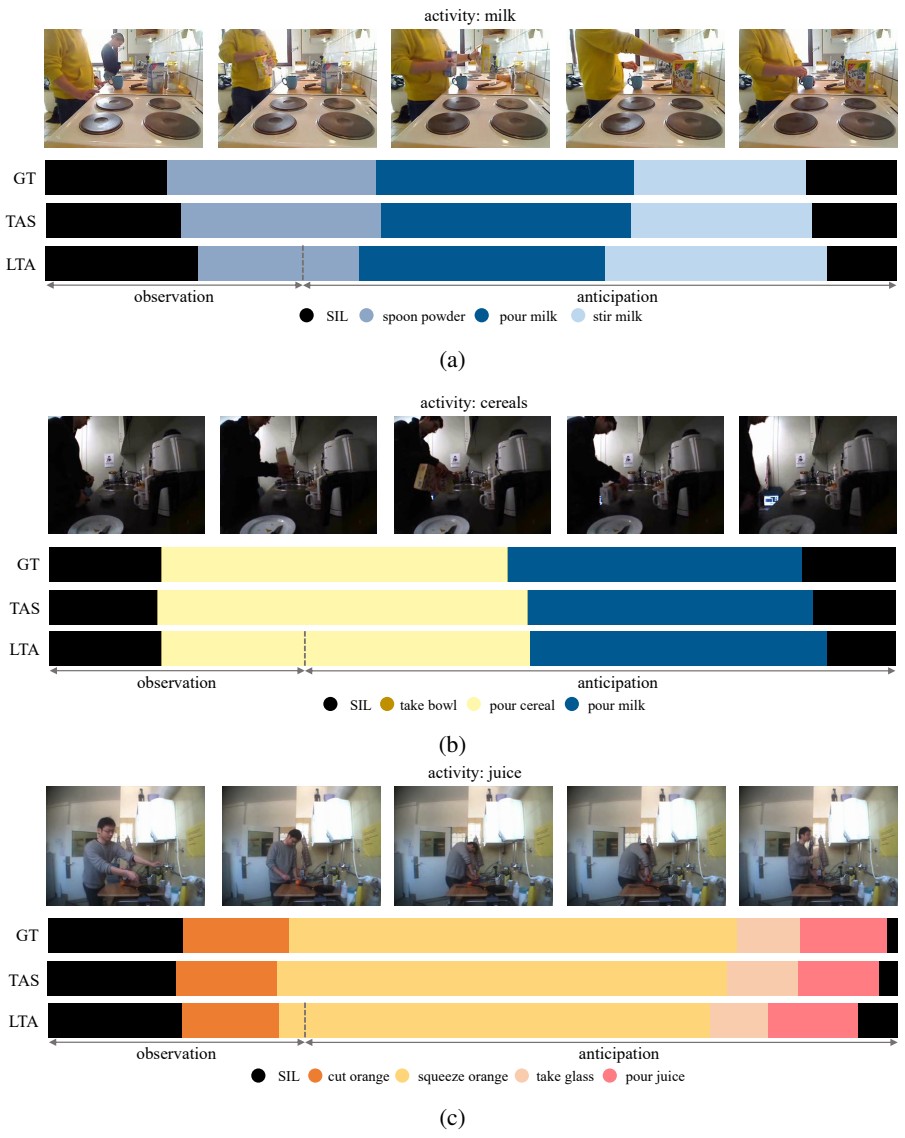

Figure S3: **Qualitative results on successful cases**

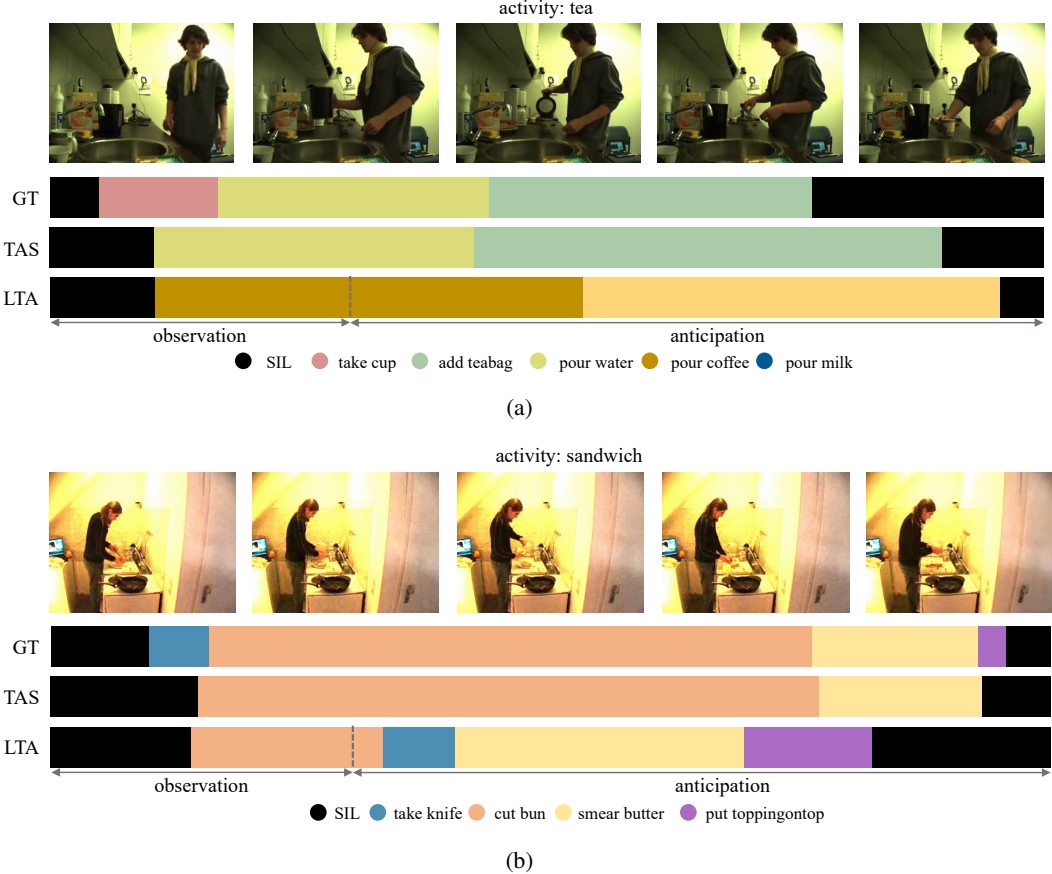

Figure S4: **Qualitative results on failure cases**

