# OpenReview forum: "ActFusion: a Unified Diffusion Model for Action Segmentation and Anticipation"
_NeurIPS.cc/2024/Conference — NeurIPS 2024 poster_

### Official Review · Reviewer_zKDZ · 2024-07-08

**Soundness:** 1
**Presentation:** 3
**Contribution:** 3
**Rating:** 6
**Confidence:** 1

**Summary:**

This paper extends DiffAct to perform both action segmentation and action anticipation. An anticipative masking with a learnable mask token is proposed. Experiments are conducted on the three common benchmark datasets.

**Strengths:**

1. The motivation for unifying action segmentation and action anticipation is reasonable, given the task similarity. It is also intuitive and reasonable to extend a generative framework from segmentation to anticipation, given the generative nature of anticipation.

2. The learnable mask token is interesting.

3. The ablation studies are relatively comprehensive.

4. Codes are provided in the supplementary.

**Weaknesses:**

1. The major technical problem is that the proposed method assumes the ground truth video length is known for action anticipation at the test time. Without the ground truth video length T, the anticipative mask M^A can not be constructed during the inference time for anticipation. This is also shown in the code provided, where 'full_len' is input into the ddim inference function. This is problematic, conflicting with the goal of anticipation, and leading to incomparable experimental results.

2. The technical novelty is limited. The main contribution is extending the DiffAct method with an anticipative mask, while other modules are from existing methods. But given the contribution of unifying the segmentation and anticipation tasks, this is not a deciding factor for me.

3. It will be better to conduct experiments on Assembly101. Given the small data size and the saturated performance on GTEA, I would recommend a transition from GTEA to Assembly101 for this task.

Minor:
- In Figure 1c, what do the triangle and the circle mean?
- In Table 1, some recent methods are missing, such as MVGA ICCV23 and RTK ICCV23.

**Questions:**

NA

**Limitations:**

The limitations were briefly discussed.

---

> ### Author Rebuttal · Authors · 2024-08-07
>
> ### **[Inference setup in LTA]**
> We would like to clarify that our model can predict future actions with an arbitrary length by adjusting the number of mask tokens needed for prediction; our model itself does not require a ground-truth length of anticipation, and the ground-truth length is used to
>  generate the same length of anticipation for the convenience of evaluation in testing. Note that previous methods [19, 24, 43] also commonly used ground-truth video lengths during inference to generate the final predictions, which can be found in the released original codes. We obtain the codes for [19] from the authors.  We hope this clarifies the reviewer's concern. Otherwise, please let us know.
>
>
> ### **[Novelty]**
> Please refer to the general response for the novelty.
>
>
> ### **[Experiments on Assembly101]**
> Due to limited resources and time, we were unable to finish the full-scale experiment on time. In response to suggestions regarding Assembly 101,  we instead did a smaller scale experiment by randomly sampling 10% of the entire training set to train the models and evaluate the performance on the full validation set to ensure a fair comparison.
>
> Table R5 compares the performance of ActFusion and LTContext [5], the state-of-the-art TAS model on Assembly 101 from the experiment. We find that our model outperforms LTContext across all metrics, showing the potential advantages of the proposed method. We will provide a full-scale experiment in the final manuscript.
>
> **[Table R5. Experiments on Assembly101]**
> | method     | F1@10 | F1@25 | F1@50 | Edit | Acc  | Avg. |
> |------------|-------|-------|-------|------|------|------|
> | LTContext [5] | 18.7  | 15.9  | 11.2  | 17.6 | 27.3 | 18.2 |
> | ActFusion (ours) | 21.7  | 19.3  | 14.0  | 19.8 | 28.1 | 20.6 |
>
> ### **[Explanation of Figure 1(c)]**
> Please refer to the general response for a detailed explanation of Figure 1(c).
>
> ### **[Missing references]**
> Thank you for letting us know. We will include MVGA [R4] and RTK [R5] in our final manuscript.
>
> [R4] B. Jiang et al. RTK: Video action segmentation via contextually refined temporal keypoints. In ICCV’23.
> [R5] N. Aziere and S. Todorovic. Markov game video augmentation for action segmentation. In ICCV’23.

---

> > ### Comment · Reviewer_zKDZ · 2024-08-08
> > **Response**
> >
> > Thanks very much for the response.
> >
> > W1: I am not convinced.
> >
> > This is not an evaluation choice as argued by the authors. This is a test data leakage issue. For any machine learning model, you should not use any ground truth when obtaining predictions on test data.
> >
> > One possible solution might be using the mean length of training videos as the 'full_len' during testing. But I guess this will lower the results.
> >
> > As for previous methods mentioned by the authors, I did not check their code. Did other previous methods except for those mentioned also use this ground truth? Even if ground truth was used in some of previous codes, I would consider this as an issue to be fixed in the following works rather than a convention to be inherited.
> >
> > W2: I appreciate the novelty of the unification, but not the methodology. This is subjective though.
> >
> > W3 and minor have been addressed.

---

> ### Author Response · Authors · 2024-08-12
>
> ### **[W1: Using the ground-truth video length during inference]**
>
> Thanks to your response, now we fully understand the point regarding the use of the ground truth video length during inference. To address this concern, we conducted additional experiments where no ground truth length is used in testing; following your suggestion, in testing, we fixed the length of future frames (i.e., mask tokens) to the maximum number of future frames in the training set.
>
> Experimental results in this setting on the 50 Salads dataset are reported in Table R6. In this table, the column ‘use of GT length’ indicates if each model exploits ground-truth length during testing. We found that, following the first paper introducing long-term dense action anticipation [2], all previous methods utilize the ground-truth length during testing, except for those whose codebases and/or inference setups are not available [25, 51, 65]; we marked these methods 'unknown' in the column.
>
> In the table, **ActFusion*** represents **our original model using a fixed number of mask tokens without retraining.** To ensure a fair comparison, we applied the same testing scheme to Farha et al [19] (denoted by Farha et al*) and FUTR [24] (denoted by FUTR*), *both of which originally utilize the ground-truth length during testing*. In this setting, ActFusion* still outperforms Farha et al* and FUTR*, and as the reviewer expected, all of these methods perform worse.
>
> This performance drop is due to the use of the ground-truth video length in training, which results in a discrepancy between the training and inference setups. To mitigate this issue, **we retrained our model while fixing the number of mask tokens to cover the maximum number of future frames in the training set.** The retrained results, presented in the last row of Table R6 and denoted as **ActFusion†**, achieve the state of the art in long-term action anticipation (LTA). We observed that fixing the number of mask tokens brings performance gain when the prediction ranges are relatively short, as the model benefits from more stable predictions. However, for longer predictions, particularly when the prediction ratio is set to 0.5, we observed performance degradation compared to ActFusion. This degradation is probably due to the fact that it becomes more difficult for the model to determine the end of an activity. Nonetheless, these results demonstrate that our method can be flexibly adapted to different numbers of mask tokens, ultimately achieving the state of the art in LTA.
>
> We sincerely hope this clarification addresses your concerns. We believe that the experimental setup you suggested will contribute significantly to the field by providing more realistic and reasonable evaluation protocols, and we will include all of the above results in the revision.
>
> **[Table R6. LTA results with and without using ground truth length]**
>
> | method            | use of GT length | $\alpha=0.2,\beta=0.1$ | $\alpha=0.2,\beta=0.2$ | $\alpha=0.2,\beta=0.3$ | $\alpha=0.2,\beta=0.5$ | $\alpha=0.3,\beta=0.1$ | $\alpha=0.3,\beta=0.2$ | $\alpha=0.3,\beta=0.3$ | $\alpha=0.3,\beta=0.5$ | Avg.  |
> |:-------------------|:-------------:|:------------------------:|:------------------------:|:------------------------:|:------------------------:|:------------------------:|:------------------------:|:------------------------:|:------------------------:|:---------:|
> | Temporal Agg. [51]| unknown| 25.50| 19.90| 18.20| 15.10| 30.60| 22.50| 19.10| 11.20| 20.26|
> | A-ACT [25]| unknown| 35.40| 29.60| 22.50| 16.10| 35.70| 25.30| 20.10| 16.30| 25.13|
> | Object Prompt [65]| unknown| 37.40| 28.90| 24.20| **18.10**| 28.00| 24.00| **24.30**| 19.30| 25.53|
> | Farha et al. [19] | ✓| 34.76| 28.41| 21.82| 15.25| 34.39| 23.70| 18.95| 15.89| 24.15|
> | Farha et al.* [19]| -| 29.07| 23.83| 20.49| 12.77| 26.51| 17.78| 14.35| 11.19| 19.50|
> | FUTR [24]| ✓| 39.55| 27.54| 23.31| 17.77| 35.15| 24.86| 24.22| 15.26| 25.96|
> | FUTR* [24]| -| 28.84| 20.01| 16.65| 11.37| 22.48| 16.49| 13.21| 9.21| 17.28|
> | ActFusion (ours)  | ✓| 39.55| 28.60| 23.61| 19.90| 42.80| 27.11| 23.48| 22.07| 28.39|
> | ActFusion* (ours) | -| 34.50| 26.17                   | 20.27                   | 11.87                   | 34.58                   | 22.75                   | 17.31                   | 11.33                   | 22.75|
> | ActFusion† (ours)| -           | **41.30**               | **30.83**               | **24.40**               | 16.10                   | **41.70**               | **28.08**               | 22.48               | **19.56**               | **28.06**|
>
> - In the table, the bolded values represent the highest accuracy among the models that do not use ground truth length, ensuring a fair comparison.

---

> ### Comment · Reviewer_zKDZ · 2024-08-12
> **Test Data Leakage and Difficult Case**
>
> Thanks very much. I really appreciate the additional experiments. It is helpful to see the new results without using ground-truth length, which are quite reasonable.
>
> It is surprising that all previous methods have utilised the ground-truth length during testing. I would flag this as a critical issue worth the community's attention. Revealing this issue using the above experiments might be a significant contribution to the community.
>
> Therefore, I have greatly increased my score, on the condition that the additional experiments will be included in the later version. And all the other experiments should also be 'totally' updated to this no ground-truth version. However, this might be too substantial and there is no mechanism to guarantee that.
>
> Overall, this is really a difficult case. Therefore, I have also changed my rating to the lowest confidence, and I would leave this to the chairs for the final judgement.

---

> > ### Author Response · Authors · 2024-08-12
> >
> > We sincerely appreciate the reviewer’s constructive feedback, which has been invaluable in raising an important issue within the community and guiding us to improve our submission. We will ensure that all experiments in our final manuscript are fully updated without the use of ground-truth length during inference.
> >
> > As the reviewer zKDZ mentioned, we also believe that revealing this critical issue and rectifying it would be a significant contribution to the community. We will thoroughly analyze and bring to light the issues shared by previous methods (at least, including [19, 24]), ensuring that all models are compared under adequate and fair conditions.

---

### Official Review · Reviewer_uA7n · 2024-07-11

**Soundness:** 3
**Presentation:** 3
**Contribution:** 3
**Rating:** 6
**Confidence:** 5

**Summary:**

This paper proposes a new unified diffusion model called ActFusion, which solves the tasks of Time Action Segmentation (TAS) and Long Term Action Prediction (LTA) in a joint learning framework. To unify the two tasks, the model effectively handles the visible and invisible parts of the sequence during the training phase; The visible part is used for observed video frames, while the invisible part is used for future expectations. The experiment showed that the model achieved state-of-the-art performance in standard benchmark tests of 50 Salad, Breakfast, and GTEA.

**Strengths:**

1. The design of the ActFusion model is novel, with its anticipative masking strategy and random masking method unifying the tasks of TAS and LTA.
2. The model enhances its performance on both tasks through mutual promotion of TAS and LTA.
3. The reported results in the paper surpass existing techniques on multiple evaluation metrics, demonstrating significant performance improvements.

**Weaknesses:**

1. What is the difference between the Diffusion model used in the paper and the DiffAct model? They appear to be similar overall, despite using different Masks to unify the tasks of TAS and LTA.
2. The paper employs so many loss functions for model training; the authors should analyze the impact of different losses rather than simply exploring the effects of the encoder and decoder.
3. Since the denoising process of the diffusion model is a time-consuming task, I am interested in the computational efficiency of the model proposed in the paper. The authors are advised to provide information on the model's GFLOPs, parameter scale, and inference time.
4. The authors need to further improve the interpretation of the figures, such as what the circles and triangles in Figure 1(c) represent.

**Questions:**

1. What is the difference between the Diffusion model used in the paper and the DiffAct model? They appear to be similar overall, despite using different Masks to unify the tasks of TAS and LTA.
2. The paper employs so many loss functions for model training; the authors should analyze the impact of different losses rather than simply exploring the effects of the encoder and decoder.
3. Since the denoising process of the diffusion model is a time-consuming task, I am interested in the computational efficiency of the model proposed in the paper. The authors are advised to provide information on the model's GFLOPs, parameter scale, and inference time.
4. The authors need to further improve the interpretation of the figures, such as what the circles and triangles in Figure 1(c) represent.

**Limitations:**

In summary, the authors have creatively used different Masks to unify the tasks of TAS and LTA with a single model and achieved significant results. However, compared to existing models, there does not seem to be much modification, highlighting a disadvantage in innovation.

---

> ### Author Rebuttal · Authors · 2024-08-07
>
> ### **[Comparison with DiffAct]**
> Please refer to the general response to the comparison with DiffAct.
>
> ### **[Loss ablation studies]**
> We conduct ablation studies on the loss functions: boundary loss, smoothing loss, and cross-entropy loss. Table R4 presents the results, demonstrating that the combination of bounding loss $L_{bd}$ and smoothing loss $L_{smo}$ is effective for both TAS and LTA. While the effectiveness of these losses in TAS is well-documented in previous research [18, 64, 42], their impact on LTA has been less explored. Notably, the smoothing loss leads to significant performance gains in both tasks, indicating that smoothed predictions are beneficial.
>
> **[Table R4. Loss ablation studies]**
>
> **(a) Results on TAS**
> | $L_{\text{bd}}$ | $L_{\text{smo}}$ | $L_{\text{ce}}$ | F1@10 | F1@25 | F1@50 | Edit | Acc  | Avg. |
> |------------|------------|------------|-------|-------|-------|------|------|------|
> |            |            | &check; | 88.4  | 86.5  | 79.1  | 82.5 | 84.9 | 84.3 |
> |            | &check; | &check; | 91.3  | 90.0  | 84.5  | 86.3 | 88.8 | 88.2 |
> | &check; | &check; | &check; | 91.6  | 90.7  | 84.8  | 86.0 | 89.3 | 88.5 |
>
> **(a) Results on LTA**
>
> | $L_{\text{bd}}$ | $L_{\text{smo}}$ | $L_{\text{ce}}$ | $\alpha=0.2, \beta=0.1$ | $\alpha=0.2, \beta=0.2$ | $\alpha=0.2, \beta=0.3$  | $\alpha=0.2, \beta=0.5$ | $\alpha=0.3, \beta=0.1$ | $\alpha=0.3, \beta=0.2$ | $\alpha=0.3, \beta=0.3$ | $\alpha=0.3, \beta=0.5$ |
> |------------|------------|------------|-------------|-------|-------|----------|-------|-------|--------|----------|
> |            |            | &check; | 35.62| 27.04 | 20.17 | 15.93| 34.38 | 22.33 | 19.96 | 16.94|
> |            | &check; | &check; | 39.19    | 28.99 | 23.13 | 19.45 | 39.53 | 25.19 | 22.67 | 19.88 |
> | &check; | &check; | &check; | 39.55 | 28.60 | 23.61 | 19.90    | 42.80 | 27.11 | 23.48 | 22.07 |
>
> ### **[Computational efficiency]**
> Table R5 compares the computational cost of our model with ASFormer [64] for TAS, and FUTR [24] for LTA, in terms of the number of parameters, GPU memory usage during inference, and inference time. For TAS, as shown in Table R5 (a), although ASFormer has fewer parameters with lower inference time, it requires more GPU memory during inference and obtains lower performance. To improve computational efficiency, we reduce the DDIM inference steps to 10 and 1. This reduction decreases inference time while maintaining superior performance over ASFormer.
>
> For LTA, as shown in Table R5 (b), our model is approximately eleven times smaller than FUTR, uses less GPU memory, but has a longer inference time. By reducing the DDIM inference steps to 1, our model achieves a similar inference time to FUTR.
> Overall, our model is practical and efficient since it can handle both TAS and LTA tasks with a unified model, eliminating the need for separate models and reducing GPU resource usage and the time required for separate training. Note that we use model checkpoints from the official GitHub repositories for all comparisons.
>
> **[Table R5. Computational efficiency]**
>
> **(a) Results on TAS**
> | method            | # inference steps | Avg. Performance | # parameters (M) | memory (GB) | inference time (s) |
> |-------------------|-------------------|------------------|------------------|-------------|--------------------|
> | AsFormer [64]         | 1                 | 81.9             | 1.134            | 0.272       | 1.66               |
> | ActFusion (ours)  | 1                 | 86.0             | 1.576            | 0.164       | 0.42               |
> | ActFusion (ours)  | 10                | 87.9             | 1.576            | 0.164       | 1.17               |
> | ActFusion (ours)  | 25                | 88.5             | 1.576            | 0.164       | 2.01               |
>
> **(a) Results on LTA**
> | method            | # inference steps | Avg. Performance | # parameters (M) | memory (GB) | inference time (s) |
> |-------------------|-------------------|------------------|------------------|-------------|--------------------|
> | FUTR [24]             | 1                 | 26.0             | 17.38            | 0.156       | 0.21               |
> | ActFusion (ours)  | 1                 | 26.2             | 1.576            | 0.151       | 0.26               |
> | ActFusion (ours)  | 10                | 27.8             | 1.576            | 0.151       | 1.04               |
> | ActFusion (ours)  | 25                | 28.4             | 1.576            | 0.151       | 2.14               |
>
> ### **[Explanation of Figure 1(c)]**
> Please refer to the general response for a detailed explanation of Figure 1(c).

---

> > ### Comment · Reviewer_uA7n · 2024-08-12
> >
> > Thank you for your detailed rebuttal. The author has clarified most of my concerns. I keep my score as weak accept.

---

> ### Author Response · Authors · 2024-08-12
>
> We thank reviewer uA7n for the motivating feedback. We are pleased to hear that most of the concerns have been addressed by our rebuttal. The results discussed in the rebuttal will be included in the final manuscript.

---

### Official Review · Reviewer_FW3D · 2024-07-24

**Soundness:** 2
**Presentation:** 3
**Contribution:** 2
**Rating:** 6
**Confidence:** 3

**Summary:**

The author introduce a unified diffusion model for temporal action segmentation (TAS) and long-term action anticipation (LTA), dubbed ActFusion, where a single model is jointly trained to address these two problems effectively. A new anticipative masking is presented for the effective unification of two tasks, along with random masking to learn intra-action relations. ActFusion achieves the state-of-the-art performance on both TAS and LTA, demonstrating the effectiveness of joint learning of two tasks across standard benchmark datasets, 50 Salads and Breakfast, and GTEA.

**Strengths:**

* Originality
    * The paper presents a approach to integrating two popular vision tasks, temporal action segmentation and long-term action anticipation, within a unified model. This is the first time these two problems have been investigated together in a single framework, highlighting the originality of the research.
* Clarity
    * The paper is clearly articulated and provides sufficient details. It offers an adequate background on diffusion model and how it is utilized. The inclusion of pseudo-code and actual code facilitates a better understanding for the reader. Additionally, the experimental settings are thoroughly described, which aids in the reproducibility of the results.

**Weaknesses:**

* The methodology lacks novelty. The model architecture, loss function (Cross-Entropy Loss, Temporal Smoothness Loss, Boundary Alignment Loss) and mask strategy (no mask, relation mask, boundary mask) closely resemble those used in the Diffusion Action Segmentation (https://arxiv.org/pdf/2303.17959v2). The primary distinction in this work is only its extension to include the long-term action anticipation task, and the introduction of anticipative masking.
* The performance improvement relative to other state-of-the-art works is marginal. On TAS task, when compared to DiffAct, this model shows an approximate 1-point improvement across various metrics (F1, edit score, frame-wise accuracy) on different benchmarks (50 Salads, Breakfast, GTEA).
* The model consistently achieves sub-optimal results when assessed using frame-wise accuracy as the metric on TAS task. As the authors point out, this could potentially be addressed by employing reconstruction methods for masked features. I am eager to see how these adjustments could enhance the model's performance.
* More ablation study for LTA task could be included, e.g., How important is past context for the models (\alpha) ? How far into the future can models predict (\beta)? Table 2 reports only a limited range of settings. A more thorough analysis on these aspects would be highly valuable.

**Questions:**

* The approach is currently limited to predicting action labels within a closed set. To extend this work to predict open-set action labels, what can be done?
* While "segmentation helps anticipation" is evident from table 3, "anticipation helps segmentation" is considerably less significant in table 4. What's the reason behind this?
* If further training the model to reconstruct the original features from the masked features, would it also improve the LTA task?
* Paper writing
    * What does circle and triangle means in figure 1(c)?
    * In table2, why there are multiple underlined values (suppose to be second-highest value) in each column?

**Limitations:**

* Limitation
    * The author notes the sub-optimal performance when using frame-wise accuracy to evaluate the TAS task and suggests a potential solution, though its effectiveness remains unproven. I am eager to see how these proposed adjustments might improve the model's performance.
* Potential negative societal impact
    * There is no potential negative societal impact of this work.

---

> ### Author Rebuttal · Authors · 2024-08-07
>
> ### **[Novelty]**
> Please refer to the general response for our novelty.
>
> ### **[Marginal performance]**
> We would like to clarify that the performance improvements achieved by our model are significant. Figure R1 in the pdf file illustrates the performance of the Top 10 TAS models for each dataset listed in Table 1, based on their average performance across all metrics. The average performance gain between the adjacent models is 0.5 percentage points (pp), 0.7 pp, and 0.6 pp for the 50 Salads, Breakfast, and GTEA datasets, respectively.
> Our model, ActFusion, achieves performance gains of 0.7pp, 0.4pp, and 1.1pp compared to the second-best models for each dataset, and 1.1pp, 0.4pp, and 1.5pp compared to DiffAct. We believe the performance gains are meaningful, with notable improvements in two datasets.
>
> ### **[Reconstruction of masked features]**
>
> Masked auto-encoding is a technique used in training NLP models like BERT [R1] and has recently been adapted to vision models [21, 27, 68]. Inspired by this approach, we train our model to reconstruct input video features from the masked tokens as an auxiliary task. Specifically, we employ MLP layers on the encoder embeddings to reconstruct the input video features and apply mean squared error (MSE) loss $L_{\text{recon}}$ during training.
>
> Table R2 shows the overall results on both TAS and LTA tasks.  In TAS, overall performance increases. We conjecture that reconstruction helps the model gain a deeper understanding of the underlying data structure and temporal dynamics by predicting the missing parts of the input. In LTA, we find that reconstruction is more effective on relatively short-term anticipation. Since short-term predictions are often based on more immediate context, there is less uncertainty. As a result, reconstructing masked features helps the model capture immediate patterns and transitions more accurately. However, for long-term predictions, as the model attempts to predict further into the future, the uncertainty increases significantly. Long-term predictions involve more variables and potential changes, making them inherently less predictable.  This increased uncertainty might cause performance degradation, making reconstruction less effective for action anticipation.
>
> **[Table R2. Effects of reconstruction loss]**
>
> **(a) Results on TAS**
> |$L_{\text{recon}}$ | F1@10 | F1@25 | F1@50 | Edit | Acc  | Avg. |
> |-|-|-|-|-|-|-|
> | -| 91.6|90.7|84.8|86.0|89.3|88.5|
> | ✓| 92.0|90.9|86.6|86.9|89.6|89.2|
>
> **(a) Results on LTA**
> |$L_{\text{recon}}$|$\alpha=0.2, \beta=0.1$ | $\alpha=0.2, \beta=0.2$ | $\alpha=0.2, \beta=0.3$  | $\alpha=0.2, \beta=0.5$ | $\alpha=0.3, \beta=0.1$ | $\alpha=0.3, \beta=0.2$ | $\alpha=0.3, \beta=0.3$ | $\alpha=0.3, \beta=0.5$ |
> |-|-|-|-|-|-|-|-|-|
> |-| 39.55| 28.60| 23.61| 19.90| 42.80| 27.11| 23.48| 22.07|
> | ✓ | 40.80| 31.02| 25.59| 13.94| 46.56| 26.22 | 18.56| 16.15|
>
> [R1] J. Devlin et al.  BERT: Pre-training of deep bidirectional transformers for language understanding. In NAACL’19.
>
> ### **[More analysis on LTA]**
> Table R3 shows the LTA performance across different observation ($\alpha$) and prediction ($\beta$) ratios. Average anticipation performance improves as the observation range increases. With more observations, uncertainty about future actions is relatively reduced, leading to more accurate predictions. Conversely, average anticipation performance decreases as the prediction range increases. Predicting further into the future presents more challenges due to increased uncertainty, as future actions become less predictable and more variable.
>
> **[Table R3. Analysis of observation and prediction ranges in LTA]**
> || $\alpha = 0.2$ | $\alpha = 0.3$ | $\alpha = 0.4$ | $\alpha = 0.5$ | $\alpha = 0.6$ | $\alpha = 0.7$ | $\alpha = 0.8$ | **Avg.** |
> |-|-|-|-|-|-|-|-|-|
> | $\beta = 0.1$ | 39.56|42.81| 29.87| 37.07| 32.74| 27.53| 36.73| **35.2** |
> | $\beta = 0.2$ | 28.6| 27.11| 25.41| 27.16| 27.68| 26.91| 42.28| **29.3** |
> | $\beta = 0.3$ | 23.61| 23.48| 22.46| 24.50| 27.99| 29.94 | -| **25.3** |
> | $\beta = 0.4$ | 22.71| 21.04| 22.44| 25.03| 31.08| -| -| **24.5** |
> | $\beta = 0.5$ | 19.9| 22.07| 22.85| 28.08| -| -| -| **23.2** |
> | $\beta = 0.6$ | 19.28| 22.78| 25.71|-|-| -| - | **22.6** |
> | $\beta = 0.7$ | 19.93 | 25.62|-|-|-|-|-|**22.8**|
> | $\beta = 0.8$ | 23.34|-|-|-|-|-|-| **23.3** |
> | **Avg.**| **24.6**| **26.4**| **24.8**| **28.4**| **29.9**| **28.1**| **39.5**||
> ### **[Extension to the open-set action recognition]**
> We appreciate the reviewer’s insight on extending our work towards open-set action recognition. To achieve this, we can use frozen image and text encoders from CLIP [R2] to obtain shared representations for actions and text embeddings. Similar to [R3], these embeddings can then be integrated into our model to enable open-set action recognition. We plan to explore this as a future direction for improving our approach.
>
> [R2] A. Radford et al. Learning transferable visual models from natural language supervision. In arXiv’21.
> [R3] D. Chatterjee et al. Opening the vocabulary of egocentric videos. In Neurips’23.
>
> ### **[Reasons: segmentation is more helpful on anticipation]**
> We find that segmentation greatly enhances anticipation, while the effect of anticipation on segmentation is less significant (L271-272). Segmentation directly improves anticipation by providing accurate contextual cues and action boundaries of the observations, enabling the model to make more precise future anticipation. In contrast, anticipation helps segmentation more indirectly. Anticipation encourages the model to consider long temporal relations of actions within an activity, which may not result in immediate performance improvement in segmentation.
>
> ### **[Explanation of Figure 1(c)]**
> Please refer to the general response for a detailed explanation of Figure 1(c).
>
> ### **[Multiple underlined values in Table 2]**
> Thank you for pointing out. The underline of the performance of DiffAct on the 50 Salads dataset will be removed.

---

> ### Author Response · Authors · 2024-08-12
> **A gentle reminder**
>
> Dear reviewer FW3D,
>
> We'd like to thank again for your effort and time dedicated to our submission. We've addressed your concerns in our rebuttal, and it would be very helpful if you could give us any further thoughts and update your scores before the author-reviewer discussion phase ends. Your opinion would be invaluable to us in improving our work, and we would be glad to respond further to your questions. Thank you for your consideration.
>
> Best regards,
> Authors

---

> > ### Comment · Reviewer_FW3D · 2024-08-12
> >
> > Thank you to the authors for providing detailed explanations and conducting additional experiments. These have addressed all of my questions and concerns. Please ensure to include these in the final version of the manuscript. I will be adjusting my rating accordingly.

---

> > > ### Author Response · Authors · 2024-08-13
> > >
> > > Thank you for the response. We are glad to hear that most of the concerns have been addressed by our rebuttal.
> > > We would like to thank reviewer FW3D once again for the insightful comments for extensive experiments and directions for our work. We make sure to include all experimental results in the final manuscript.

---

### Author Rebuttal · Authors · 2024-08-07

We thank all the reviewers for their insightful comments and suggestions. We are happy to see that the reviewers have given our work a positive evaluation, noting that “this is the first time these two problems have been investigated together in a single framework, highlighting the originality (FW3D)”, “the design of the AcFusion is novel (uA7n)”, “the model enhances its performance on both tasks through mutual promotion of TAS and LTA (uA7n)” and “it is also intuitive and reasonable to extend a generative framework from segmentation to anticipation, given the generative nature of anticipation (zKDz)”.

Nevertheless, the reviewers also point out important comments that:
1. the novelty of the proposed method should be explained,
2. revealing the effects of learning reconstruction of masked features for TAS is suggested,
3. further analyses on LTA, loss functions, computational cost, and dataset are suggested.

Through this rebuttal, we aim to clearly expose our novelty and provide further experimental results and analyses. We will revise the manuscript by incorporating the detailed comments from the reviewers.

In the general response, we address the questions posed by all reviewers regarding novelty and the explanation of Figure 1(c).

### **[Novelty]**
The primary novelty of the proposed method lies in unifying the two popular video tasks, temporal action segmentation (TAS) and long-term action anticipation (LTA). **The unification is not merely an extension but a novel framework that leverages the bi-directional benefits between TAS and LTA, maximizing synergies between these tasks.** None of the previous work [42, 24, 51] neither has introduced a single unified model to tackle the two tasks nor explored the bi-directional benefits. This integration is indeed crucial for practical applications, such as human-assistant robots, which need to recognize and anticipate future actions simultaneously.

**To achieve successful task integration, we introduce two types of masking strategies: anticipative masking ($M^\texttt{A}$) and random masking ($M^\texttt{R}$).** Anticipative masking plays a crucial role in effective task integration and random masking leverages to learn intra-action relations from a video. **However, simply incorporating these masking strategies does not necessarily guarantee the optimal performance for both TAS and LTA.** In Table R1, we applied anticipative and random masking to DiffAct by replacing visual embeddings with zero vectors after encoder processing and using them as conditions for the diffusion process in the decoder.
The results in Table R1 show that these masking strategies improve TAS performance but remain below existing state-of-the-art models for LTA [51, 19, 25, 24, 65],

We hypothesize that this is likely due to the limited information from the zero vectors used in future anticipation, which does not fully leverage the information from the visible tokens. To address this, **we propose a learnable masking strategy, where input visual features are replaced with learnable mask tokens provided to the encoder.** These tokens are trained to learn temporal relations between visible and invisible parts through attention mechanisms. **In our model, both the encoder and decoder are trained to handle visible and invisible parts for effective task unification.** The introduced masking strategy maximizes synergies between the two tasks, leading to achieving state-of-the-art performance in both TAS and LTA in Tables 1 and 2.  We believe our approach presents a novel integrative framework for unifying the two tasks by introducing an effective learnable masking strategy with two types of masking.

**[Table R1. Effects of a learnable masking strategy]**

(a) **Results on TAS**

| method   | $M^\texttt{A}$ | $M^\texttt{R}$ | F1@10 | F1@25 | F1@50 | Edit | Acc  | Avg. |
|---------|-------|-------|-------|-------|-------|------|------|------|
| DiffACT | -     | -     | 90.1  | 89.2  | 83.7  | 85.0 | 88.9 | 87.6 |
| DiffACT | ✓     | ✓     | 91.1  | 89.8  | 84.1  | 85.9 | 88.9 | 87.9 |
| ActFusion (ours)    | ✓     | ✓     | 91.6  | 90.7  | 84.8  | 86.0 | 89.3 | 88.5 |

(a) **Results on LTA**

| method   | $M^\texttt{A}$ | $M^\texttt{R}$ | $\alpha = 0.2$ $\beta = 0.1$ | $\alpha = 0.2$ $\beta = 0.2$ | $\alpha = 0.2$ $\beta = 0.3$ | $\alpha = 0.2$ $\beta = 0.5$ | $\alpha = 0.3$ $\beta = 0.1$ | $\alpha = 0.3$ $\beta = 0.2$ | $\alpha = 0.3$ $\beta = 0.3$ | $\alpha = 0.3$ $\beta = 0.5$ |
|---------|-------|-------|-----------------------------|-----------------------------|-----------------------------|-----------------------------|-----------------------------|-----------------------------|-----------------------------|-----------------------------|
| DiffACT | -     | -     | 11.8                        | 11.3                        | 11.3                        | 10.7                        | 20                          | 17.2                        | 16.5                        | 16.6                        |
| DiffACT | ✓     | ✓     | 30.3                        | 27.0                        | 19.1                        | 11.3                        | 37.4                        | 22.1                        | 15.6                        | 13.0                        |
| ActFusion (ours)   | ✓     | ✓     | 42.8                        | 33.9                        | 26.0                        | 20.7                        | 43.1                        | 25.8                        | 21.3                        | 20.7                        |

### **[Explanation of Figure1(c)]**

We apologize for not providing detailed explanations for Figure 1(c). In this figure, the circles represent the main tasks the models are proposed to address, while the triangles indicate auxiliary tasks used during training but not evaluated. We will include the descriptions in the final manuscript.

---

### Decision · Program_Chairs · 2024-09-25

**Decision:**

Accept (poster)

**Comment:**

All 3 reviewers were inclined to accept the paper. All reviewers agree the paper tackles an important problem in an interesting way and the results show that the method improves over previous SOTA significantly. The AC has read all reviews and, aligned with the reviewer recommendation, the AC is recommending the paper be accepted to NeurIPs.